# Crystal scavenging from mush piles recorded by melt inclusions

Penny E. Wieser [1]*, Marie Edmonds [1], John Maclennan[1], Frances E. Jenner[2] & Barbara E. Kunz [2]

Olivine-hosted melt inclusions are commonly used to determine pre-eruptive storage conditions. However, this approach relies on the assumption that co-erupted olivines have a simple association with their carrier melts. We show that primitive olivine crystal cargoes and their melt inclusions display a high degree of geochemical disequilibrium with their carrier melts at Kīlauea Volcano, Hawai'i. Within a given eruption, melt inclusions trapped in primitive olivine crystals exhibit compositional diversity exceeding that in erupted lava compositions since 1790 CE. This demonstrates that erupting liquids scavenge crystal cargoes from mush piles accumulating diverse melt inclusion populations over timescales of centuries or longer. Entrainment of hot primitive olivines into cooler, evolved carrier melts drives post-entrapment crystallization and sequestration of $CO_2$ into vapour bubbles, producing spurious barometric estimates. While scavenged melt inclusion records may not be suitable for the investigation of eruption-specific processes, they record timescales of crystal storage and remobilization within magmatic mush piles.

[1] Department of Earth Sciences, University of Cambridge, Downing Street, Cambridge CB2 3EQ, UK. [2] School of Environment, Earth and Ecosystem Sciences, The Open University, Walton Hall, Milton Keynes, Buckinghamshire MK7 6AA, UK. *email: pew26@cam.ac.uk

The utility of melt inclusions, pockets of silicate melt trapped in growing crystals, is well established in igneous petrology and volcanology. Olivine-hosted melt inclusions are commonly used to investigate pre-eruptive processes in basaltic volcanoes, including magma storage depths[1], magma mixing[2] and the factors controlling eruption style[3,4]. However, it is becoming increasingly apparent that melt inclusions are not a perfect archive of magmatic processes occurring at depth. The post-entrapment crystallisation (PEC) of olivine on the walls of the melt inclusion during cooling of the host crystal with progressive fractional crystallisation, or upon eruption, changes the major and trace element composition of the remaining melt[5,6]: the concentration of elements that are compatible in olivine decrease (e.g., MgO, Ni), while the concentration of incompatible elements increase (e.g., Nb, La, Sm, $H_2O$, $CO_2$)[7]. These changes, combined with a drop in inclusion pressure, favour the formation of a $CO_2$-rich vapour bubble[6–8]. Unless the $CO_2$ content of the bubble is quantified (e.g., using Raman spectroscopy), melt inclusion analyses using techniques such as secondary ion mass spectrometry (SIMS) or fourier transform infra red spectroscopy (FTIR) will underestimate the $CO_2$ content at the time of entrapment[8–10]. Furthermore, global compilations of melt inclusion data demonstrates that the process of decrepitation, where the host olivine ruptures and releases $CO_2$ due to a large pressure difference between the inclusion and the host melt, accounts for the significantly lower entrapment pressures recorded by melt inclusions than independent petrological barometers (e.g., clinopyroxene-liquid)[7].

Rapid diffusion rates of $H^+$ in olivine mean that melt inclusion water contents are vulnerable to diffusional re-equilibration[11]. This process may produce anomalously low water contents if the sample is not quenched rapidly upon cooling (allowing the melt inclusion to equilibrate with the degassed carrier melt)[12,13], or anomalously high water contents due to entrainment into a water-rich carrier melt, or the mixing of compositionally diverse melts[14]. A recent study at Llaima Volcano, combining melt inclusion volatile data with diffusive modelling of major element zoning in host olivines, demonstrated that melt inclusions record the progressive mixing of melts stored at various levels in the plumbing system for months to years prior to their eventual eruption[15].

In addition to the processes discussed above, which alter melt inclusion geochemistry pre- and post-entrapment, the increasingly prevalent view of magmatic systems as mush-dominated[16] raises more fundamental questions regarding the utility of melt inclusions. Settled crystals may be stored at a wide range of depths within extensive crustal cumulate piles for many millenia[16–18]. The re-entrainment of these crystals into unrelated magma batches challenges the common assumption that melt inclusions and matrix glasses (the solidified carrier melt) are related[19], such that inclusions provide a record of the pre-eruptive storage and evolution of the erupted melt. Instead, a significant proportion of erupted crystals may be antecrysts; commonly defined as crystals which formed in a separate magma batch to the one in which they were erupted[1,17,20]. Here we assess crystal-melt relationships using olivine-hosted melt inclusions from Kīlauea Volcano, Hawai'i, to assess the utility of melt inclusion records in a mush-dominated volcanic system.

Kīlauea Volcano provides the ideal test site to assess the influence of long-lived mush piles on melt inclusion records. Kīlauea is one of the most extensivelystudied volcanoes in the world; geophysical monitoring data collected over the last century, combined with numerous geochemical studies, have led to a well-constrained model of the plumbing system. Primitive basaltic magmas supplied from the Hawaiian hotspot at >100 km depth[21] ascend through the lithosphere into two main crustal

storage reservoirs situated beneath the summit of Kīlauea[22–24]. Geophysical observations indicate that the deeper, South Caldera (SC) reservoir is located at ~2–6 km depth[22,25], while the shallower Halema'uma'u (HMM) reservoir is located at ~1 km depth[24]. The presence of two distinct mixing trends in Pb isotope ratios of lavas erupted since the 1970s corroborates geophysical evidence that magma is stored in two main reservoirs[26,27]. A combination of geophysical and geochemical observations suggests that the SC reservoir supplies magma to extra-caldera and rift zone eruptions[26,27], while the HMM reservoir supplies intra-caldera summit eruptions and summit lava lakes[22,25].

It is well accepted that extensive olivine mush piles are located in close proximity to magma storage reservoirs. For example, localised regions with P-wave velocities of 6.5–7 km/s (indicative of dunitic cumulate bodies) are located at ~5 km depth beneath the summit, close to the inferred base of the SC reservoir[28]. Mass balance constraints indicate that these bodies form through the settling of olivine from fractionating melts. Primary melts have >16 wt%[29,30] MgO, yet the average bulk-rock composition of erupted melts is ~10 wt% MgO. This difference in MgO contents requires the removal of >14 vol% olivine between the intrusion of primary melts into the plumbing system, and their eruption at the surface[31]. In just a century, ~1.4 $km^3$ of olivine must accumulate from fractionating melts (for magma supply rates of ~0.1 $km^3$/yr)[23].

Erupted products from Kīlauea Volcano exhibit prominent changes in primary melt composition over decades and centuries, offering a means of fingerprinting specific time periods. Ratios of elements with similar incompatibility during crystal fractionation (e.g., Nb/Y, La/Yb) and isotopic ratios (e.g., $^{206}Pb/^{204}Pb$, $^{87}Sr/^{86}Sr$) show pronounced changes over decadal to centurial timescales, resulting from heterogeneity in the mantle source[32], conditions of melting[33], and incomplete melt mixing during magma ascent and storage[26,27,34]. For example, Nb/Y increases from ~0.4 to 0.7 between ~1800 and 1930 CE, before falling again to ~0.49 in 1982[35]. Concurrently, $^{206}Pb/^{204}Pb$ rises from ~18.40 to ~18.65 and back to ~18.40[36].

The presence of extensive olivine mush piles provides numerous opportunities for carrier melts to entrain crystal cargoes that formed in previous magma batches, while changes in melt chemistry with time allows assessment of crystal-melt relationships and crystal residence times. Melt inclusions trapped within olivine phenocrysts should track the changes in carrier melt compositions with time. However, melt inclusions hosted within olivine 'antecrysts' plucked from mush piles by an intruding magma should record the compositional range of historic magma batches in which these olivines crystallised, rather than the carrier melt in which they were erupted[1,2]. The relationship between melt inclusions and carrier melts may also be assessed using experimentally-determined Fe-Mg partitioning between olivine and melt[30,37]. Comparison of olivine forsterite contents, where Fo = 100 X $MgO_{(mol\%)}/(MgO_{(mol\%)} + FeO_{(mol\%)})$, to the equilibrium olivine composition that would crystallise from their carrier melts provides a first assessment of the degree of major element equilibrium. The forsterite contents of phenocrystic olivines should lie close to the equilibrium olivine composition, except in the case of delayed fractionation[38] or prolonged storage in close proximity to their host melts[39]. In contrast, antecrystic olivines may show deviations from equilibrium substantially exceeding the experimental uncertainty in olivine-liquid $K_D^{Fe^{2+}-Mg}$.

To investigate the degree of equilibrium between erupted melts and their crystal cargoes, we analysed melt inclusions, host olivine crystals and matrix glasses (for analytical details see Methods) from four eruptions temporally associated with activity at Mauna Ulu on the upper East Rift Zone (ERZ) of Kīlauea (Fig. 1). These tephra

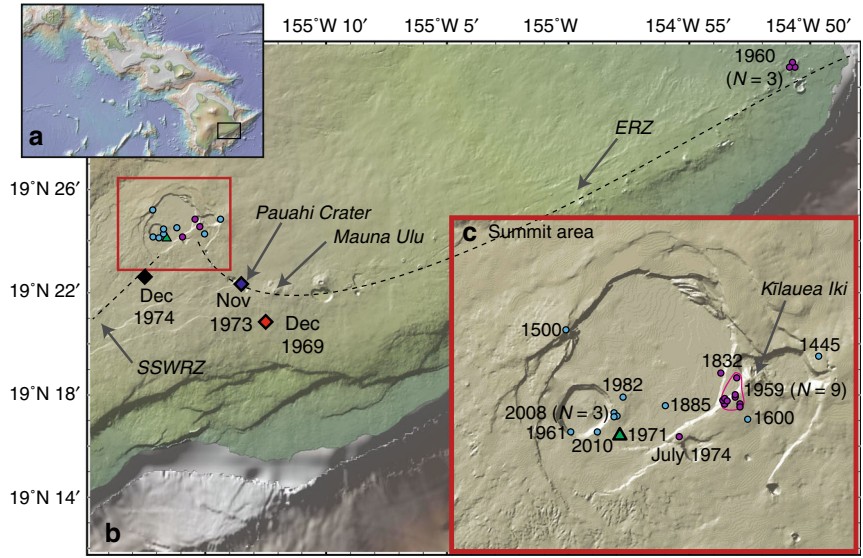

**Fig. 1** Sample locations examined in this study and the literature. **a** Map of the Hawaiian Island Chain, with the black rectangle showing the location of Kīlauea Volcano on the island of Hawaiʻi (expanded in part **b**). **b** Map of Kīlauea Volcano showing the locations of eruptions analysed in this study (bold text), alongside eruptions in the literature with melt inclusion and glass trace element data. Multiple episodes were examined from the 1959, 1960 and 2008 eruptions. Kīlauea's East Rift Zone (ERZ) and Seismic South West Rift Zone (SSWRZ) are shown in dashed black lines. **c** An expanded map of Kīlauea's summit area (red rectangle in **b**). Maps were produced in GeoMapApp (ref. [64]; http://www.geomapapp.org), and show the topography of Kīlauea before the onset of the 2018 summit collapse. Diamond symbols (red, blue and black) denote eruptions analysed in this study with mean forsterite contents >$Fo_{84}$, while triangular symbols (green) denote the 1971 eruption analysed in this study (mean forsterite content <$Fo_{84}$). Literature eruptions with mean forsterite contents >$Fo_{84}$ and <$Fo_{84}$ are shown as magenta and cyan circles respectively (see Table 1).

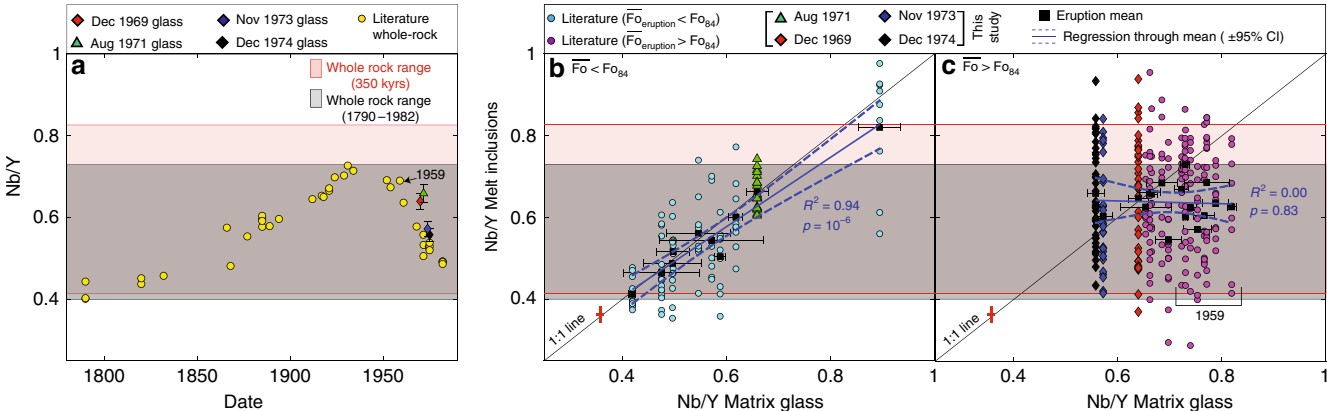

**Fig. 2** Trace elements in melt inclusions hosted by primitive olivines record melt compositional variations over centuries to millennia. **a** Whole-rock data from summit eruptions displays large temporal variations in Nb/Y (refs. [35,36]). Matrix glasses from the four eruptions analysed in this study follow similar trends, although the 1971 intracaldera eruption lags behind the May 1969 ERZ eruption (error bars show ±1σ for glass measurements). The grey box shows the range of Nb/Y ratios in whole-rock compositions since 1790 (ref. [36]), and the pink box shows the range of Nb/Y ratios in whole-rock compositions over the last 350 kyrs (ref. [44]). **b**, **c** Nb/Y ratios of melt inclusions versus co-erupted matrix glasses (±1σ glass analyses). Regression lines were calculated for the mean glass and melt inclusion composition of each eruption (95% confidence intervals on the regression are shown with dashed lines). Eruptions with mean forsterite contents <$Fo_{84}$ display a strong correlation between the mean Nb/Y ratio of melt inclusions and matrix glasses (**b**). While the 1961 summit eruption (glass Nb/Y ~0.89) certainly strengthens the observed correlation, the regression is still very good if this eruption is excluded ($R^2$ = 0.84, $p = 10^{-4}$). In contrast, there is no correlation between the mean Nb/Y ratios of melt inclusions and matrix glasses from eruptions with mean forsterite contents >$Fo_{84}$ (**c**). In particular, almost all melt inclusions from various episodes of the 1959 Kīlauea Iki eruption have significantly lower Nb/Y ratios than the composition of co-erupted matrix glasses. Nb/Y is shown for consistency with previous studies (refs. [1,27,36]), and because this ratio shows the largest percent variation in historic lavas (see Supplementary Table 6). Additionally, this ratio is not sensitive to the fractionation of olivine and chromite, or post-entrapment crystallisation (see Supplementary Figs. 5 and 6). The red cross on **b**, **c** shows the precision for repeated analysis of BCR-2 (±1σ) in melt inclusions and glasses.

and spatter samples were erupted during the highest fountaining phase of the Mauna Ulu eruption in December, 1969 (ERZ), the fissure eruption of August, 1971 (intra-caldera), the Pauahi Crater eruption of November, 1973 (ERZ), and the fissure eruption of December, 1974 (Seismic South West Rift Zone; SSWRZ[24]).

The 5-year period over which our samples were erupted includes some of the most rapid historic changes in melt composition at Kīlauea (Fig. 2a). We supplement our dataset of 126 melt inclusions and 40 matrix glass chips with literature studies where trace elements were reported in ≥8 inclusions, and

**Table 1 Key data on the samples analysed in this study, and those compiled from the literature.**

| | Date | Mean Fo (mol %) | Location | Reference | # of MI |
|---|---|---|---|---|---|
| $\overline{\text{Fo}} > \text{Fo}_{84}$ | 1832 | 86.0 | Extracaldera | Sides et al. (3) | 9 |
| | 1959 (episodes 1, 2, 3, 5, 6, 7, 8, 15, 16) | 87.5, 86.8, 86.1, 87.2, 87.1, 85.9, 86.8, 86.4, 86.0 | Extracaldera | Sides et al. (3) | 10, 11, 13, 8, 9, 12, 10, 10, 15 |
| | 1960 | 85.7 | ERZ | Sides et al. (3) | 17 |
| | 1960 (2 episodes) | 85.5, 88.4 | ERZ | Tuohy et al. (1) | 10, 19 |
| | Dec 1969 | 86.7 | ERZ | This study | 37 |
| | Nov, 1973 | 85.6 | ERZ | This study | 27 |
| | July, 1974 | 85.4 | Extracaldera | Sides et al. (3) | 8 |
| | Dec, 1974 | 87.3 | SSWRZ | This study | 42 |
| | | | | | |
| $\overline{\text{Fo}} < \text{Fo}_{84}$ | 1445 | 80.6 | Intracaldera | Sides et al. (3) | 12 |
| | 1500 | 83.3 | Intracaldera | Sides et al. (3) | 9 |
| | 1885 | 81.5 | Intracaldera | Sides et al. (3) | 10 |
| | 1961 | 82.4 | Intracaldera | Sides et al. (3) | 9 |
| | Aug, 1971 | 82.8 | Intracaldera | This study | 20 |
| | 1982 | 82.2 | Intracaldera | Sides et al. (3) | 9 |
| | 2008 (3 episodes) | 82.6, 82.6, 82.9 | Intracaldera | Sides et al. (3) | 9, 20, 9 |
| | 2010 | 81.7 | Intracaldera | Sides et al. (3) | 10 |

The symbol next to each row, and the row color, shows the classification of eruptions used in Figs. 1–4. Date: date of eruption for each sample (CE). Nine episodes of the Kīlauea Iki eruption, and 2 episodes of the 1960 Kapoho eruption (samples Kap6 and Kap 8 from ref. 1) were used. Mean Fo (mol %) mean forsterite content of olivines in each sample (calculated as 100 X $MgO_{mol\%}$/ ($FeO_{mol\%}$ + $MgO_{mol\%}$); Location: site of eruption (ERZ – East Rift Zone, SSWRZ – Seismic South West Rift Zone). Reference with the number referring to the list of references at the end of this article

co-erupted matrix glasses (Table 1). The combined dataset of 27 eruptive episodes and 384 melt inclusions spans ~600 years of eruptive history at Kīlauea, and incorporates large variations in matrix glass (Fig. 2b, c) and whole-rock compositions (Fig. 2a)[35]. This dataset reveals that melt inclusions hosted by primitive olivine crystals show trace element diversity comparable to that observed in a record of erupted liquid compositions spanning several centuries, but no clear affinity to the composition of their carrier liquids. Combined with considerable major element disequilibrium, this demonstrates that melt inclusion archives at Kīlauea are obscured by the accumulation and storage of inclusion-bearing crystals within magmatic mush piles, followed by their remobilisation and transport to the surface by unrelated carrier melts.

## Results and discussion

**Constraints from olivine-liquid disequilibrium.** Olivine forsterite contents in Kīlauean magmas show a strongly bimodal distribution[40] (Fig. 3b), which is readily apparent in a compilation of 2810 erupted olivine core compositions ($N = 689$ from our analysis, $N = 2121$ from the literature, see Methods). The two distinct peaks are centred at ~$Fo_{87-88}$ and ~$Fo_{80-82}$, with a dearth of olivines with ~$Fo_{84}$ (Fig. 3b). Within a given eruption, olivine forsterite contents fall mostly within one peak (Fig. 3a), permitting classification of eruptions based on mean forsterite contents ($\overline{\text{Fo}}$) into those with primitive ($\overline{\text{Fo}} > \text{Fo}_{84}$) and evolved crystal cargoes ($\overline{\text{Fo}} < \text{Fo}_{84}$).

Primitive crystal cargoes were observed in the three rift eruptions analysed in this study (1969, 1973, 1974; red, blue and black diamonds in Fig. 3a), and in 14 eruptive episodes from the literature (1832 and July 1974 eruption, 9 episodes of the 1959 Kīlauea Iki eruption, and 3 episodes of the 1960 Kapoho eruption; magenta circles in Fig. 3a; Table 1)[3,19]. Evolved olivine compositions are observed in the intra-caldera eruption of 1971 (this study; green triangles in Fig. 3a), and 9 eruptive episodes from the literature (1445, 1500, 1885, 1961, 1982 and 2010 eruptions, and 3 episodes of the 2008 summit eruption, cyan circles in Fig. 3a; Table 1)[3]. Primitive crystal cargoes are significantly out of major element equilibrium with their matrix

glasses, while evolved crystal cargoes lie close to the equilibrium composition (Fig. 3a).

In addition to exhibiting bimodal forsterite contents, lavas erupted around Kīlauea's summit also show bimodal whole-rock MgO contents[40], centred at ~7 ± 1 wt% and ~10 ± 2 wt% MgO[27]. This bimodality has been attributed to magma storage in two distinct reservoirs, as eruptions between 1969 and 1982 thought to have tapped the HMM reservoir based on Pb isotopes[26,27] have whole-rock MgO contents aligned with the evolved peak, while extra-caldera and rift eruptions thought to have tapped the SC reservoir align with the primitive peak[27]. The spatial distribution of olivine compositions is similar[40]; historic and prehistoric intra-caldera eruption products host evolved olivines ($\overline{\text{Fo}} < \text{Fo}_{84}$) while extra-caldera and rift eruption products host more primitive olivines ($\overline{\text{Fo}} > \text{Fo}_{84}$; Figs. 1 and 3). Thus, we suggest that bimodality in olivine forsterite contents is also generated, in part, by fractionation within two separate reservoirs, as progressive fractional crystallisation within a single reservoir would produce steadily declining amounts of olivine with a given forsterite content between ~$Fo_{90-81}$ (red line, Fig. 3b). Although the geometry of magma storage at Kīlauea may undergo changes following caldera-forming events (e.g., 1924, 2018)[26,41], the fact that prehistoric tephra and lava show a similar, bimodal distribution of glass and olivine compositions suggests that variably evolved melts were stored at two distinct depths over the time period evaluated in this study[27,40].

A fractional crystallisation model over the suggested range of melt compositions present in the HMM reservoir (~7 ± 1 wt% MgO)[27] produces olivine compositions overlapping with the low forsterite olivine peak (red dashed line in grey box; Fig. 3b). This, combined with the high degree of major element equilibrium between olivine crystals and their matrix glasses (Fig. 3a), implies that these crystals grew in equilibrium with their carrier liquids. The narrow range of evolved liquid (and therefore olivine) compositions in the HMM reservoir[26] likely results from an increase in the enthalpy of crystallisation, and a decrease in the rate of change of Mg# for a given amount of crystallisation as clinopyroxene joins the liquidus assemblage[27,42]. However, fractional crystallisation over the range of melt compositions erupted from the SC reservoir (~10 ± 2 wt% MgO)[27] produces a

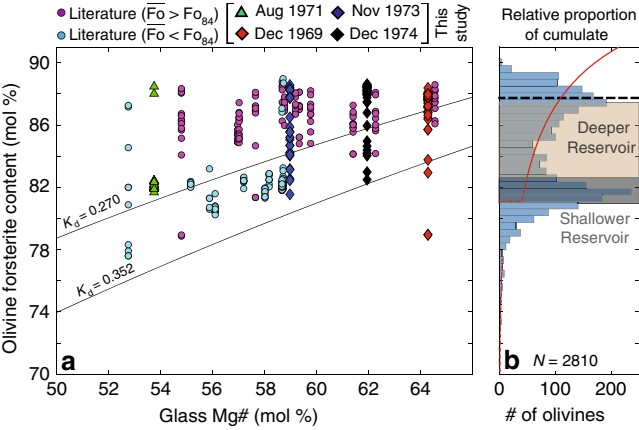

**Fig. 3** Primitive crystal cargoes consist largely of antecrysts. **a** Olivine forsterite contents versus matrix glass Mg# ($Fe^{3+}/Fe_T = 0.15$; ref. [65]). Olivine-liquid equilibrium lines are shown for $K_D = 0.270$–$0.352$ (refs. [30,37]). Eruptions with evolved crystal cargoes (mean forsterite content <$Fo_{84}$; cyan circles and green triangles) contain olivines lying closer to olivine-liquid equilibrium lines than eruptions with primitive crystal cargoes (mean forsterite content >$Fo_{84}$; magenta circles, blue, black and red diamonds). **b** Compilation of 2810 Kīlauean olivine core compositions (this study and literature; see Methods). The red line shows the expected distribution of forsterite contents following fractional crystallisation within a single reservoir from a primary melt composition (17.1 wt% MgO; ref. [29]; see Methods). The solid section of the line represents the composition of olivines that would crystallise from primary melts, down to the composition of the most evolved melts present within the SC reservoir. The dashed section of the line represents the olivine compositions that would crystallise from more evolved melts. The beige box shows the range of olivine compositions that would crystallise from the observed distribution of glass compositions in eruptions thought to originate in the SC reservoir (~10 ± 2 wt% MgO), while the grey box shows the olivine compositions that would crystallise from the observed distribution of glass compositions in eruptions thought to originate in the shallower, HMM reservoir (~7 ± 1 wt% MgO; ref. [27]). The black dashed line shows the mean olivine composition produced by the fractionation trajectory indicated by the solid red line (17.1 wt% MgO to 8 wt% MgO). Models run in Petrolog3 at QFM, $K_D = 0.3$ (see Methods).

distribution of olivine forsterite contents which steadily declines between ~$Fo_{87.4-82.6}$ (section of red line within beige box; Fig. 3b). Crucially, compared with the observed distribution of forsterite contents, the modelled distribution shows no peak, and is shifted to substantially lower forsterite contents (Fig. 3b). Producing the peaked distribution of primitive olivine compositions with fractional crystallisation alone is problematic. Unlike the evolved HMM reservoir, it is difficult to envisage an efficient thermal or chemical buffering process operating in the small-volume (0.1–0.3 km³)[26] SC reservoir fractionating only olivine and chromite.

Similar peaks at high forsterite contents in Icelandic lavas have been attributed to diffusive re-equilibration of Fe and Mg within crystal mush piles, which produces a peaked distribution of forsterite contents centred on the mean composition of the mush pile[39]. Fractionation from estimated primary melt compositions (~17.1 wt% MgO)[29] to the most evolved liquid composition in the SC reservoir (~8 wt% MgO; solid section of red line)[27] produces an olivine mush pile with a mean olivine composition of ~$Fo_{87.7}$ (black dashed line; Fig. 3b). Thus, the random scavenging of olivines stored in diffusively re-equilibrating mush piles by subsequent carrier melts accounts for the high degree of major

element disequilibrium (Fig. 3a), and the strongly peaked distribution of primitive forsterite contents in many different eruptions (Fig. 3b).

Assessing storage timescales from peaked forsterite distributions requires knowledge of the initial distribution, and the mush pile height. Thomson and Maclennan[39] explored this parameter space for Icelandic lavas assuming that a sill of variable thickness underwent progressive fractional crystallisation in a closed system. However, the SC reservoir is an open system, with primitive melts entering at the base, and variably evolved melts leaving the reservoir to be erupted, or stored within the rift zones[43]. Repeated injection of more primitive melts likely produces cyclic variations in forsterite content with height. Clearly, an alternative approach is required to assess the residence times of primitive olivine crystals in open (and therefore highly unconstrained) systems such as Kīlauea.

**Constraints from melt inclusion-liquid disequilibrium.** To test the hypothesis that evolved olivines are phenocrysts forming within the HMM reservoir, while primitive olivines represent antecrysts scavenged from extensive mush piles at the base of the SC reservoir, we compare the trace element compositions of matrix glasses and olivine-hosted melt inclusions. Phenocrystic olivines from the HMM reservoir should contain melt inclusions with trace element signatures overlapping those of their carrier melts (matrix glasses), while antecrystic olivines from the SC reservoir should host melt inclusions with no obvious geochemical affinity to their carrier melts.

Matrix glasses from the four eruptions investigated in this study have distinct trace element signatures (Supplementary Figs. 1 and 2a). Matrix glass trace element ratios which are relatively insensitive to crystal fractionation of predominantly olivine (and minor chromite), such as Nb/Y, follow similar temporal trends to whole-rock measurements from summit lavas (Fig. 2a)[35,36]. Within a given eruption hosting evolved crystal cargoes (Table 1), melt inclusion Nb/Y ratios show limited chemical diversity, centred on the composition of the co-erupted matrix glass (Fig. 2b). This supports our hypothesis that evolved crystal cargoes consist predominantly of phenocrysts, which grew in the HMM reservoir from the melt which carried them to the site of the eruption.

In contrast, melt inclusions from the three rift eruptions investigated in this study (1969, 1973, 1974) are highly diverse, with indistinguishable mean Nb/Y ratios despite statistically significant changes in the mean matrix glass composition (1969 glasses are distinct from 1973–1974 glasses; ANOVA, $\alpha = 0.05$; Fig. 2a, c). In fact, of the 17 eruptions with primitive crystal cargoes (Table 1), only two of the melt inclusion populations have distinguishable means at $\alpha = 0.05$. A linear regression through the mean melt inclusion and matrix glass composition for eruptions with primitive crystal cargoes shows no trend ($R^2 = 0.00$), unlike the excellent correlation observed for eruptions with evolved crystal cargoes ($R^2 = 0.94$; Fig. 2b, c).

Within a single eruption hosting a primitive crystal cargo, melt inclusion heterogeneity exceeds the range of Nb/Y ratios observed in historic summit lavas (1790–1982)[36], and a record of erupted lavas spanning 350 kyrs[44] (Fig. 2). Similarly, it has been demonstrated for several Hawaiian volcanoes that Pb isotopes in melt inclusions from a single eruption show diversity comparable to the total observed variation at a given volcano[45]. Extreme melt inclusion variability in basaltic lavas is commonly attributed to the presence of a diverse range of mantle melts within the plumbing system at any given time[2,46]. Basaltic lavas from Iceland and mid oceanic ridges show substantial reductions in trace element variability with forsterite content, tracking the

concurrent mixing and fractionation of diverse mantle melts[2,46,47]. In contrast, Kīlauean melt inclusions exhibit no obvious correlation between trace element diversity and olivine forsterite contents (Supplementary Fig. 2). Additionally, the presence of remarkably coherent temporal variations in lava geochemistry at widely-spaced eruption sites at Kīlauea suggests that eruptions are tapping a well-mixed reservoir[27,42]. Melt homogeneity at Kīlauea has been attributed to efficient mixing in storage reservoirs between primary and resident magmas[26,48]. Alternatively, the combination of thick lithosphere and high mantle potential temperatures beneath Hawai'i may permit extensive mixing during long ascent paths from the mantle. In particular, supraliquidus ascent may homogenise melts prior to the crystallisation of olivine and the entrapment of melt inclusions[34,49,50].

Regardless of the exact mechanism producing the relatively homogenous reservoir compositions, the apparent absence of diverse melt compositions within the plumbing system based on erupted lava compositions implies that diverse melt inclusion populations were acquired from many different, well-mixed reservoir compositions present in the plumbing system over prolonged periods. An approximate estimate of the contribution to the diversity of melt inclusions from reservoir heterogeneity and magma mixing can be obtained from the range of trace element ratios in melt inclusions from the 1971 summit eruption (Fig. 2b; Nb/Y = 0.6–0.74). This accounts for only 25% of the variation in Nb/Y ratios observed in the 1969 eruption (Fig. 2c; Nb/Y = 0.37–0.94). Thus, indistinguishable melt inclusion populations with a broad range of melt inclusion trace element ratios in many different primitive eruptions, combined with the lack of relationship with forsterite contents, supports a model in which carrier melts randomly scavenge olivine antecrysts from mush piles containing highly diverse melt inclusion populations just prior to eruption[2].

**Centurial storage times within mush piles.** An estimate of the minimum residence times of primitive olivine crystals in mush piles can be obtained by comparing the range of trace element ratios in melt inclusions to erupted lava compositions. Bulk-rock and glass analyses from the 1959 Kīlauea Iki eruption lie close to upper limit of Nb/Y ratios observed since ~1790 (Fig. 2a). Yet, the vast majority of co-erupted melt inclusions have significantly lower Nb/Y ratios (Fig. 2c), down to ~0.4. Melts with these compositions were only present in Kīlauea's plumbing system before ~1790 (Fig. 2a), suggesting that these erupted crystals were stored for at least 170 years. It is highly possible that crystals were stored for considerably longer timescales; the range of Nb/Y ratios in melt inclusion records from the three rift eruptions investigated in this study greatly exceed the range of bulk-rock compositions between 1790 and 1982. In fact, the 1969 eruption displays trace element diversity surpassing the range of erupted lava compositions over a 350-kyr period (Fig. 2c).

Our study is the first to recognise centurial storage timescales of crystal mushes at Kīlauea. Most previous estimates of timescales are based on the interdiffusion of Fe and Mg within olivine[51,52]. The relatively high diffusion coefficients of these species means that profiles within individual crystals only record the final processes happening decades to hours before their eventual eruption[51,52]. Diffusion times reported from modelling of trace element zoning in deformed Kīlauean olivines yield slightly longer timescales (<33–238 days for P, and 10 days to 43 years for Cr)[53]. However, the highly disparate timescales obtained from these two trace elements highlights the difficulty in obtaining storage estimates from olivine trace element contents, due to the uncertainty in experimentally determined diffusivities

(ref. [54] vs. ref. [55]). In particular, the presence of sharp, P zones in slowly cooled igneous intrusions suggests that P diffusion may be considerably slower than available experiments, so the storage times of Bradshaw et al.[53] are very much minimum estimates.

A related question is whether trace elements within olivine-hosted melt inclusions undergo diffusive equilibration with their surrounding melts during centurial storage. While early estimates of diffusion rates for rare earth elements suggest that diffusive re-equilibration may occur over tens to hundreds of years[54], more recent studies calculate diffusivities that are ~3 orders of magnitude lower (requiring $10^4$–$10^6$ years for 50% re-equilibration of Ce and Yb in a 50 μm melt inclusion within a ~1 mm olivine at 1300 °C)[55]. Evidence for the lack of trace element re-equilibration in our dataset is provided by the similarity of regression lines for incompatible elements defined by matrix glasses and melt inclusions (e.g., Nb vs. La, Nb vs. Ce; Supplementary Fig. 3). These strong correlations are likely produced by different extents of mantle melting[33]. If trace element re-equilibration was occurring within melt inclusion populations during prolonged mush pile storage, different extents of re-equilibration for different trace elements (and different inclusion and host olivine sizes) would result in melt inclusions defining more scattered correlations with different regression lines compared with matrix glasses. While there are no available experimental estimates for the diffusivities of Nb and Y in olivine, the similar correlations defined by melt inclusions and matrix glasses for Nb vs. Ce and Y and Yb (Supplementary Fig. 3) suggests that Nb and Y are also resistant to diffusive re-equilibration during centurial storage. Thus, comparisons of melt inclusion diversity to erupted melt compositions provides the most reliable estimate of storage timescales in long-lived mush-dominated systems.

**Disequilibrium affects melt inclusion CO₂ systematics.** The entrainment of hot, primitive antecrysts into cooler, more evolved carrier melts generates thermal disequilibrium, in addition to the major and trace element disequilibrium discussed above. The characteristic conductive cooling time ($\tau$) of an olivine with a radius ($l$) of 0.5 mm and a thermal diffusivity ($\kappa$) of $5.6 \times 10^{-7}$ m²/s[56] is ~0.5 seconds ($\tau = l^2/\kappa$)[57]. In contrast, the characteristic diffusional time scale for forsterite contents is ~80 years ($\tau = l^2/D$, where $D_{Fo} \sim 10^{-16}$)[39]. Thus, hot primitive antecrysts (and their melt inclusions) reach thermal equilibrium with cooler, more evolved carrier melts long before major element equilibrium is achieved. Rapid cooling drives post-entrapment crystallisation on the inclusion wall. The efficiency of this process is demonstrated by the similarities between MgO contents (which are a proxy for temperature)[58] of melt inclusions and co-erupted glasses in eruptions with $\overline{Fo} > Fo_{84}$ (Supplementary Fig. 4), despite the strong disequilibrium that still exists between olivine forsterite contents and matrix glass Mg#s (Fig. 3a). As the MgO contents of the melt inclusions have rapidly re-equilibrated following entrainment, the degree of olivine-melt disequilibrium (calculated by subtracting the equilibrium forsterite content for the carrier melt from the measured forsterite content) is the best proxy for the temperature difference between entrained olivines and host melts. This parameter strongly correlates with the amount of PEC calculated in Petrolog3 (Fig. 4a). The maximum amount of PEC is experienced by the most primitive (and hottest) olivines which are entrained into the most evolved (and coolest) melts. Our hypothesis that post-entrapment crystallisation is a direct result of the rapid, thermal re-equilibration following the scavenging of primitive olivines into evolved carrier melts[19] differs from the common view that PEC is driven by cooling during progressive fractional crystallisation[5].

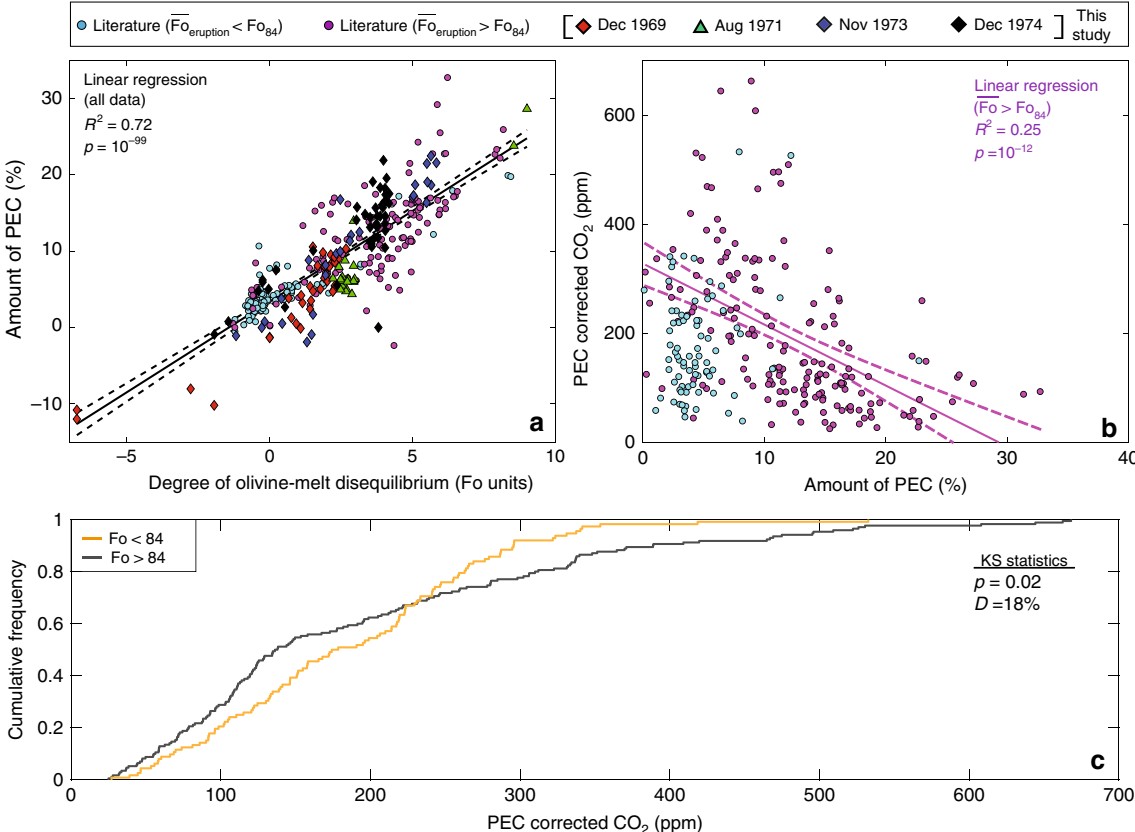

**Fig. 4** Post-entrapment crystallisation of inclusions in primitive olivines causes significant $CO_2$ sequestration into shrinkage bubbles. **a** Degree of olivine-melt disequilibrium versus the amount of post-entrapment crystallisation (PEC). Olivine-melt disequilibrium calculated by subtracting the equilibrium forsterite content of the erupted matrix glass ($K_D = 0.3$, $Fe^{3+}/Fe_T = 0.15$; ref. [65]) from the olivine forsterite content. A linear regression through all data points is represented by a black line, with 95% confidence intervals indicated by dashed black lines. **b** Amount of PEC versus melt inclusion $CO_2$ concentrations[19,73]. A linear regression through melt inclusions from eruptions with mean forsterite contents >$Fo_{84}$ is represented by a purple line, with 95% confidence intervals indicated by dashed purple lines. Amount of PEC calculated in Petrolog3 (ref. [66]). **c** Cumulative frequency plot of $CO_2$ concentrations in melt inclusions from ref. [3] subdivided into those hosted by olivines with <$Fo_{84}$ and >$Fo_{84}$. The test statistics and p-values from two sample Kolmogorov-Smirnov (KS) tests are shown. To increase the number of datapoints on panels **b**, **c** (as no $CO_2$ data was collected for the samples of this study), all eruptions in ref. [3] were used, rather than just those with ≥8 melt inclusions.

Melt inclusions in eruptions with primitive crystal cargoes are hosted within olivine crystals which are significantly out of major element and thermal equilibrium with their carrier melts, and as a result, have experienced large amounts of PEC (up to 30%). It is well established that changes in major element chemistry, combined with a drop in inclusion pressure with increasing PEC, favour $CO_2$ saturation and the formation of a $CO_2$-rich vapour bubble[6–8]. This mechanism accounts for the clear decrease in melt inclusion $CO_2$ concentrations with increasing amounts of PEC[19] in these eruptions (Fig. 4b). Thus, $CO_2$ contents measured by SIMS or FTIR in melt inclusions (which frequently contain vapour bubbles)[3] hosted within entrained antecrystic olivines are far lower than the concentration at the point of entrapment[1,8]. Vapour bubble formation following crystal scavenging can explain the lack of correlation between forsterite content and $CO_2$ contents at Kīlauea[1]. For example, 70% of olivines with >$Fo_{84}$ record lower pressures than more evolved olivines (<$Fo_{84}$; Fig. 4c). This is at odds with conceptual models of the plumbing system, where more primitive melts crystallise at deeper pressures[1,27,40]. In fact, 70% of inclusions hosted in olivines with >$Fo_{84}$ record $CO_2$ concentrations indicating entrapment at <2 km depth[3,59], which is significantly shallower than the geophysical estimates for the depth of the SC

reservoir[22,24]. Bubble growth driven by PEC means that reliable barometric estimates can only be gained from analysis protocols accounting for the amount of $CO_2$ held within the melt inclusion (SIMS or FTIR) and the vapour bubble (e.g., Raman spectroscopy)[9]. Such PEC-driven bubble growth is particularly problematic at Kīlauea, where the absence of clinopyroxene and plagioclase in most erupted lavas precludes the use of other petrological barometers.

In contrast, the concentration of $CO_2$ in melt inclusions hosted within evolved crystal cargoes indicate entrapment pressures between ~8–75 MPa[3], with most inclusions clustering between 25–50 MPa[3]. These pressures encompass geophysical constraints on the depth of the HMM reservoir (pressures of ~25–50 MPa for storage depths of 1–2 km[24] and densities of ~2600 kg/m$^3$[43]). Thus, melt inclusion $CO_2$ contents in evolved crystal cargoes produce reliable barometric estimates, even though these inclusions contain bubbles. This implies that these bubbles only contain a small fraction of the total $CO_2$ budget. $CO_2$-poor bubbles may form during post-eruptive cooling, due to differences between the glass transition temperature and the temperature at which C-diffusion becomes extremely slow (allowing bubble growth, but hindering the diffusion of $CO_2$ from the melt into the bubble)[7,60,61].

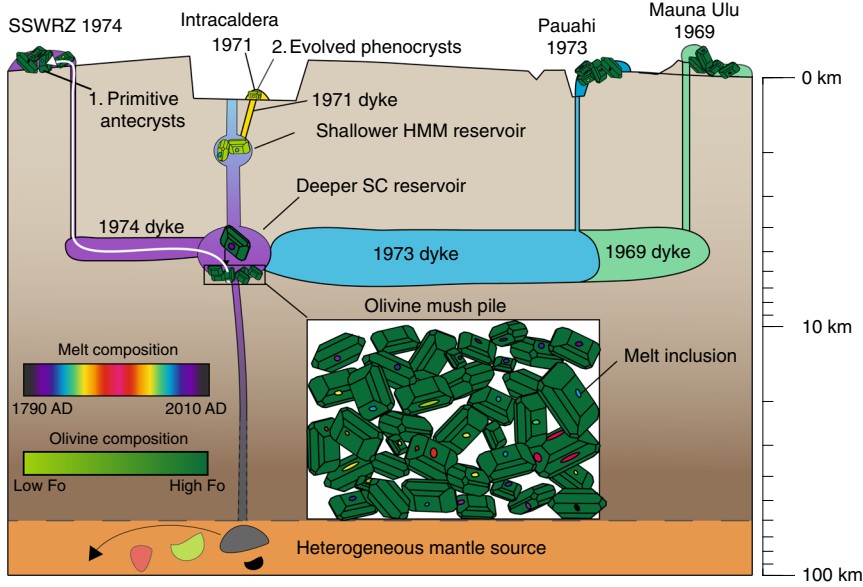

**Fig. 5** Kīlauea's upper and lower reservoir contain distinct populations of olivines, with corresponding variations in melt inclusion diversity. Diverse melt compositions are produced by a heterogenous mantle source (indicated on this diagram by the different coloured blobs in the mantle producing chemically diverse melts feeding Kīlauea's plumbing system; refs. [26,27,35,36]). Mixing processes on ascent and during reservoir replenishment produce a near homogeneous reservoir composition which changes progressively with time (black–red–black over ~200 yrs, Fig. 2a). At a given time, relatively homogeneous reservoir compositions produce melt inclusions with a narrow range of trace element compositions. Olivines (containing melt inclusions) settle to form extensive mush piles at the base of the deeper, SC reservoir (ref. [67]). Rift zone and extracaldera eruptions are supplied by dykes propagating from the SC reservoir. These erupted liquids scavenge antecrysts from mush piles just prior to eruption, and acquire a melt inclusion record which sampled changing reservoir compositions over centuries to millennia, hosted in a chemically unrelated carrier liquid (**1**). In contrast, intracaldera eruptions (e.g., August 1971) are supplied by dykes propagating from the upper HMM reservoir (refs. [26,27]). These dykes transport phenocrystic olivines to the surface along with the carrier melts in which these olivines crystallised (**2**). Magma transport to each eruption site is shown as a dyke tapping the composition of the reservoir at that point in time (after ref. [27]). Note that the vertical scale is schematic.

**Pre-eruptive history of evolved crystal cargoes**. The HMM reservoir, supplying intra-caldera eruptions, is thought to be replenished from the SC reservoir (supplying extra-caldera and rift eruptions[27]; Fig. 5). This geometry explains the lag in Nb/Y ratios in the August 1971 intra-caldera eruption products relative to the December 1969 ERZ eruption (Fig. 2a). Erupted olivine and melt compositions from the HMM reservoir show restricted major element diversity, due to thermal and chemical buffering at the onset of clinopyroxene fractionation. The evolved crystal cargoes ($\overline{Fo} < Fo_{84}$) in intra-caldera summit eruptions consist predominantly of phenocrysts, shown by the high degree of olivine-melt Fe-Mg equilibrium (Fig. 3a) and overlapping melt inclusion and matrix glass trace element ratios (Fig. 2b). Low crystal contents of intra-caldera eruptions, together with similar whole-rock and glass MgO contents, provide further evidence that melts from the HMM reservoir have not scavenged significant quantities of antecrystic olivines[42]. Thus, melt inclusion populations in eruptions hosting evolved crystal cargoes provide a reliable record of the pre-eruptive storage and evolution of their host melts. The high degree of major element equilibrium between olivine and glass compositions limits the amount of PEC experienced by melt inclusions (<10%; Fig. 3b), resulting in $CO_2$ entrapment pressures consistent with geophysical constraints on the depths of the HMM reservoir.

**Pre-eruptive history of primitive crystal cargoes**. The composition of parental melts supplying the SC reservoir changes with time due to variations in mantle source composition and melting conditions[26,27,35,36] (Fig. 5). Efficient mixing during ascent from the mantle, and/or following reservoir replenishment, creates a magma batch with limited compositional heterogeneity at any

given time[34,35,48]. The injection of hotter mantle melts into the cooler reservoir may create conditions favouring the formation of melt inclusions[62], which trap the restricted range of melt compositions present in the reservoir at any given time. The typical MgO contents of primitive rift zone and extra-caldera lavas (~10 wt% MgO)[27] requires substantial fractionation of olivine[31] from high MgO mantle melts (~15–17 wt% MgO)[29]. Over centuries, crystal fractionation combined with rapidly changing reservoir compositions produces a mush pile of olivines hosting chemically diverse melt inclusions. Diffusive re-equilibration of Fe–Mg within the mush pile over centuries (or longer) creates a prominent peak in olivine forsterite contents at ~$Fo_{87.7}$, while slower diffusion rates for incompatible trace elements preserve melt inclusion diversity[46].

Prior to eruption, carrier melts scavenge mush-pile olivines (and their diverse melt inclusion populations), explaining the similarity in melt inclusion records in many different eruptions, and the pervasive major and trace element disequilibria between crystal cargoes and their co-erupted glasses. Entrainment of hot, primitive olivines into lower temperature carrier melts drives large amounts of PEC, creating a melt inclusion record which yields spuriously low entrapment pressures from $H_2O–CO_2$ barometry. Scavenging of mush-pile olivines with a restricted range of forsterite contents due to diffusive re-equilibrium into carrier melts can account for the approximately linear olivine addition trends in whole-rock compositions at Kīlauea (projecting back to $Fo_{86-87.5}$)[40,63]. The linearity of these trends has previously been explained by delayed fractionation[38]. However, this explanation is inconsistent with the peaked distribution of primitive olivine compositions, and the large degree of trace element disequilibrium between melt inclusions and matrix glasses (Fig. 2c). Overall, these findings challenge the common

assumption that the most primitive crystals provide the most pristine record of pre-eruptive processing.

**Mush piles convolute melt inclusion records**. The compositional relationships between melt inclusions, host olivines, and co-erupted carrier liquids at Kīlauea Volcano reveals that primitive crystal cargoes resided in mush piles for centuries before their eventual eruption in chemically unrelated carrier liquids. The mush pile likely resides at the base of the South Caldera magma reservoir (~3–5 km depth). Primitive olivine crystals trap melt inclusions from this reservoir, which experiences cyclic variations in magma chemistry, and settle into extensive mush piles. Melt inclusions erupted in 1959, 1969, 1973, and 1974 exhibit Nb/Y ratios that were last observed in erupted lavas in ~1790; providing evidence that crystals were stored for almost two centuries (and possibly much longer) before their eruption. Only more evolved olivines, which formed within the shallower (~1 km) HMM reservoir, are true phenocrysts providing a record of the processes that occurred in the days to weeks prior to the eruption of a given magma batch.

The history of pre-eruptive processes preserved in melt inclusions can be obscured by accumulation and storage of inclusion-bearing crystals in mushes, followed by crystal scavenging by new carrier liquids. This decoupling of inclusions and crystals from their carrier liquids may be commonplace at volcanoes where high melt fluxes create long-lasting magma reservoirs and associated mush piles. While the processes acting to degrade the melt inclusion record following entrapment are well recognised, we demonstrate that the common assumption that melt inclusions record pre-eruptive processes is flawed in mush-rich systems. However, scavenged melt inclusion records are far from redundant; they provide novel insights into the storage, and subsequent remobilisation, of crystals in magmatic mush piles.

## Methods

**Analytical methodology**. Spatter and tephra samples from the four Mauna Ulu period eruptions (1969, 1971, 1973, 1974) were jaw crushed, then sieved into 3 size fractions (250–840 μm, 840–1000 μm, > 1000 μm). Olivines were picked and individually mounted in crystalbond^TM on a glass slide. Individual melt inclusions were exposed by grinding with 250–1500 grade wet and dry paper. This method allowed embayments (melt inclusions with a connection to the outside of the crystal) to be identified and discarded. Care was taken to analyse inclusions hosted within olivine crystals from all three size fractions. Olivines containing melt inclusions were mounted in epoxy in groups of ~40, and polished with progressively finer silicon pastes. Samples were carbon coated prior to EPMA analysis. We analyse 37, 27, 42 and 20 inclusions respectively from the 1969, 1973, 1974 and 1971 eruptions (see Table 1), and ~10 matrix glass chips from each eruption.

Major elements in matrix glasses, melt inclusions and host olivines were measured using a Cameca SX100 EPMA in the Department of Earth Sciences, University of Cambridge. Olivines containing melt inclusions were analysed in a dual condition set up. Si, Mg and Fe were analysed in the first condition, at 15 kV, 20 nA and a 1 μm beam size. Al, Ti, Ca, Ni, Cr and Mn were analysed in the second condition, at 15 kV, 100 nA and a 1 μm beam size. Count times and calibration materials, alongside estimates of precision and accuracy based on repeated measurements of a San Carlos Olivine secondary standard, are shown in Supplementary Table 1. Glass analysis was conducted with a dual condition set up. Na, Al, P, Ca, K, Ti, Si, Mg, Fe and Mn were analysed in the first condition, using 15 kV, 10 nA with a beam size of 10 μm. S and Cl were analysed in the second condition on both the PET and LPET crystal at 15 kV, 40 nA, and a beam size of 10 μm. Count times and calibration materials, alongside estimates of precision and accuracy are shown in Supplementary Table 2 and 3. Precision and accuracy were calculated from repeated measurements on VG2 and A99 secondary standards. Small melt inclusions (<30 μm diameter) were analysed one at a time to ensure that the beam was in the centre of the melt inclusions.

Trace elements analysis in matrix glasses and melt inclusions was performed using a Photon Machines G2 193 nm excimer laser system, equipped with a HelEx II 2-volume cell coupled to an Agilent 8800 ICP-MS/MS at the School of Environment, Earth and Ecosystem Sciences at The Open University. Analyses were conducted following the techniques described by ref. [68]. Run conditions utilised a 10-Hz repetition rate; a fluency of 3.63 J/cm$^2$ on the sample surface, and ablation cell gas of 0.91 min$^{-1}$ He. For spot sizes of 25 μm, 5 ml min$^{-1}$ N$_2$ was

| Table 2 Kīlauean eruptions with published melt inclusion trace element data. | |
| --- | --- |
| **Reference** | **Eruptions used** |
| Sides et al.[3] | 1445, 1500, 1832, 1885B, 1959 (Ep 1, 2, 3, 5, 6, 7, 8, 15, 16), 1960, 1961, 1974-J2, 1982, 2008 (X 3), 2010, |
| Tuohy et al.[1] | 1960 Kapoho (Kap6 and Kap8). |

added downstream to increase sensitivity. Matrix glasses were analysed with a spot size of 110 μm. Melt inclusions were analysed at varying spot sizes (65 μm, 50 μm, 40 μm, 30 μm and 25 μm) depending on the diameter of the inclusion (care was taken to avoid vapour bubbles). Melt inclusions were analysed manually to ensure that the laser spot was placed such that it did not overlap with the host olivine. Ni signals were monitored during each analysis to determine whether the laser ablated inclusions or underlying olivine. Signals were carefully selected in Iolite to only include the signal from the melt inclusion. Background was measured for 30 s prior each analysis, followed by 30 s of signal and 50 s washout. NIST-SRM 612 was used for external calibration and $^{43}$Ca for internal calibration of trace element data. BCR-2G was used as the secondary standard to monitor precision and accuracy. For phosphorus measurements, BCR-2G was used for external calibration of data, with $^{29}$Si for internal calibration. Melt inclusions were analysed in batches consisting of a standard bracket of two NIST 612 spots, 3 BCR-2G spots analysed before and after ~15 individual melt inclusions. Glasses were analysed in a single batch, with 2 BCR and 2 NIST612 spots at the start, middle and end of the run. Standard values are reported in Supplementary Data 1, and compared to Open University preferred values for BCR–2 G. Almost all measurements lie within ±5% of the preferred values.

**Olivine compositional database**. Olivine compositions shown in Fig. 3b were compiled from a variety of sources. A GEOROC search (Jan 2018) was conducted for all shield stage olivines from Kīlauea. Analyses specified as rims were discarded. As this database was compiled to gain insight into magma storage at depth, olivines from lava lakes and experimental studies were discarded. Finally, any olivines without stated locations were discarded, in case they represented lake/experimental olivines. This dataset was supplemented with publications missing from GEOROC[3,19,51,69,70]. Additionally, 689 olivine core compositions from 12 different eruptions distributed across the Kīlauean edifice (Supplementary Table 4) were determined by EPMA in three separate analytical sessions. Details of the various analytical conditions, and estimates of precision and accuracy is provided in Supplementary Table 5. The resulting database of forsterite contents, and references for GEOROC olivine compositions is provided (Supplementary Data 2). Complete compositional data is presented in ref. [67].

**Literature melt inclusion trace element data**. We compile all Kīlauean eruptions from the literature where trace element data is reported in 8 or more inclusions, and in the co-erupted matrix glass (Table 2).

**Petrolog modelling**. Fractional crystallisation models in Petrolog3[66] were ran using the primary melt composition of Clague et al.[29] at 1 kbar, QFM (using the model of ref. [71]). The FeO$_T$ for the primary melt composition was increased by a factor of 3.2% to provide a better match with glass data. Mineral melt models of ref. [72] were used for olivine, clinopyroxene and plagioclase. Olivine $K_D$ was set at 0.3. The amount of PEC was calculated using the Petrolog3 "Olivine MI" tool at QFM. Host olivine compositions were taken from EPMA analysis. FeO* values were set at 11.33 wt% based on observed liquid lines of descent (and for consistency with previous studies)[19].

## Data availability

All compositional data obtained in this study are included as supplementary data tables.

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

## Acknowledgements

We thank Giulio Lampronti and Iris Buisman for help collecting SEM maps and EPMA data. Isobel Sides (funded by a Natural Environment Research Council [NERC] studentship) and Don Swanson (U.S. Geological Survey) collected the samples used in this study. P.E.W. is funded by NERC DTP studentship NE/L002507/1. B.E.K. is funded by NERC grant NE/P017045/1.

## Author contributions

P.E.W. conceived the project with M.E. and J.M.; P.E.W. prepared and analysed the melt inclusions by EPMA, and LA-ICP-MS under the guidance of F.J. and B.E.K.; P.E.W. interpreted the data and wrote the manuscript with help from all authors.

## Competing interests

The authors declare no competing interests.
