## [Peer Review File · Nature Communications]

Reviewers' comments:

Reviewer #1 (Remarks to the Author):

This well-written manuscript relating the compositions of melt inclusions, matrix glass, and host olivines presents an interesting model for magma storage and mobilisation for the Kilauea system. While I have some minor comments regarding the data presentation and handling, the main issue I have is whether the findings advance our understanding of magmatic systems conceptually.

I would like to summarise this concern by highlighting two statements in the abstract:

"Entrainment of primitive olivines into more evolved carrier melts causes crystallisation on the inclusion walls and the sequestration of CO₂ into vapour bubbles, producing spurious barometric estimates."

"Thus, the provenance of the melt inclusion record must be carefully considered before this archive is related to eruption-specific measures."

Both of these are presented as major findings and I don't see how this is new compared to what we already know. E.g. A recent article by Ruth et al. published in Nature communications shows that magma storage, mobilisation and assembly can be complex especially when considering the integration of melt inclusions and host mineral chemistry. While they don't specifically focus on mush piles in their paper, they definitely address the difficulties of 'carefully' considering the melt inclusion record. This is an example from an arc setting, however, the overall implications are the same.

The former statement about the need to account for CO₂ in bubbles and walls is even more well established in multiple papers (e.g.: Aster et al. 2016, Esposito et al. 2016, Moore et al. 2015). So, I am not sure that with the exception of an interesting mush pile model for Kilauea what else is conceptually novel.

Minor aspects:

It is unclear to me why new datasets were generated and supplemented with literature data. At the end most of the key results are largely on the basis of literature data (e.g. Figure 3b).

I am also wondering about what the reference millenia-style storage is based on (fig. 2 and text references). The vast majority of the variations in Nb/Y is accommodated in the last <100 years (Fig 2a). So, the entire dynamics could still be reflecting that timeframe not requiring a long millenia scale mush (while I personally could see it existed, the data is not showing that).

If you think the changes in CO₂ are related to post-entrapment crystallisation and that Nb/Y is not affected by it, while incompatible element concentrations are getting enriched, it would be interesting to see this relationship between Nb and PEC or Y and PEC.

How important is the high Nb/Y glass matrix sample to account for the excellent R² in figure 2b. While that relationship is still there without this subset of data, it seems to be heavily weighted towards it.

How much do you have to worry about diffusion effects on Nb and Y in the melt inclusions through diffusion in the olivine? While Cottrell et al (2002) suggested that incompatible elements should be safe, they also stated: "Examination of published trace element concentrations of olivine hosted inclusions show little evidence for reequilibration, at least for the light REE and other highly incompatible elements. It is difficult, however, to provide firm constraints due to the uncertainties in olivine diffusivities and the initial condition." Diffusivities of trace elements in olivine has been a major debate in the last few years starting with Spandler and O'Neill (2010). If diffusivities in olivine are much greater than initially thought (especially for high Si activity systems) then fidelity

of the trace element record in the melt inclusions may also be compromised, especially for millennia-old mush pile melt inclusions.

References:

- Moore LR, Gazel E, Tuohy R, Lloyd AS, Esposito R, Steele-MacInnis M, Hauri EH, Wallace PJ, Plank T, Bodnar RJ. Bubbles matter: An assessment of the contribution of vapor bubbles to melt inclusion volatile budgets. *American Mineralogist*. 2015 Apr 1;100(4):806-23.
- Esposito R, Lamadrid HM, Redi D, Steele-MacInnis M, Bodnar RJ, Manning CE, De Vivo B, Cannatelli C, Lima A. Detection of liquid H₂O in vapor bubbles in reheated melt inclusions: Implications for magmatic fluid composition and volatile budgets of magmas?. *American Mineralogist*. 2016 Jul 1;101(7):1691-5.
- Aster EM, Wallace PJ, Moore LR, Watkins J, Gazel E, Bodnar RJ. Reconstructing CO₂ concentrations in basaltic melt inclusions using Raman analysis of vapor bubbles. *Journal of Volcanology and Geothermal Research*. 2016 Sep 1;323:148-62.
- Ruth, D.C., Costa, F., de Maisonrouve, C.B., Franco, L., Cortés, J.A. and Calder, E.S., 2018. Crystal and melt inclusion timescales reveal the evolution of magma migration before eruption. *Nature communications*, 9(1), p.2657.
- Cottrell E, Spiegelman M, Langmuir CH. Consequences of diffusive reequilibration for the interpretation of melt inclusions. *Geochemistry, Geophysics, Geosystems*. 2002 Apr 1;3(4):1-26.
- Spandler C, O'Neill HS. Diffusion and partition coefficients of minor and trace elements in San Carlos olivine at 1,300 C with some geochemical implications. *Contributions to Mineralogy and Petrology*. 2010 Jun 1;159(6):791-818.

Reviewer #2 (Remarks to the Author):

This paper presents an interesting and novel study of the relationship between melt inclusion bearing crystals and their carrier melts at Kilauea volcano. It provides important context for studies using melt inclusions (from Hawaii and elsewhere) and shows in a simple but convincing way that the inclusions in the more primitive crystals are probably the least faithful recorders of pre-eruptive conditions. This is contrary to what is generally assumed. It is well written and the data looks to be of high quality. I have no major concerns that would be a barrier to publication. The data presented certainly supports the conclusions.

Prior to publication, however, they may want to consider adding some additional discussion on some of the unexplored aspects of their data/model. For example, why do the higher Fo olivines trap (on-average) more enriched melts than the lower Fo crystals? Is the implication that all MIs that have compositions that differ from the carrier melt are antecrysts? Or do they see a role for some of the MIs in recording the melt mixing process? While the mean of the lower Fo MIs matches that of the carrier melt, the range of values from several of the eruptions covers most of the 350 kyr range in Nb/Y. What do these outlier values represent?

Reviewer #3 (Remarks to the Author):

Key Results/Manuscript Summary:

The manuscript titled, "Crystal scavenging from mush piles recorded by melt inclusions," provides an evaluation of new and literature-derived in-situ geochemical analysis of glasses, melt inclusions, and olivines from the tephra and lava of historic eruptions from Kilauea Volcano in

Hawaii. The data analysis of magmas from numerous eruptions identifies two main olivine populations: a lower Fo olivine population which hosts melt inclusions derived from a relatively shallow, magmatic reservoir with melt inclusions related to the matrix glasses and a higher Fo populations (more primitive) that hosts compositional heterogeneous melt inclusions populations, which are likely stored in deeper reservoirs (despite barometric estimates from CO₂/H₂O, which the authors suggest are underestimated because of post-entrapment processes), that are not in equilibrium with their host glasses. These results are used to hypothesize a model for a storage and ascent of Hawaiian magmas that involves scavenging of crystal cargo from more deeply stored reservoirs with heterogeneous melt inclusion populations. The manuscript warns that the presence of 'antecrysts' is a complexity that is imperative to identify in any studies utilizing melt inclusions.

General Comments:

While this manuscript provides an interesting case study and uniquely combines a variety of recently established hypotheses in the field of melt inclusion geochemistry, I do not feel that the manuscript adheres to the four main criteria for publication in a Nature Research journal. The manuscript is well-written and the data and analysis provide strong support for the presented conclusions, but it does not strike me to be "of extreme importance to scientists" in my field likely not "interesting to researchers in other related disciplines." This is actually highlighted by the manuscript itself which relies heavily on previous interpretations to provide a framework for their study. For example, evidence for 2 separate storage reservoirs at Kilauea was first suggested based on Pb isotope distinctions (ref 15), and forsterite composition peaks previously attributed to crystal mush pile processes in Iceland (ref 21). Additionally, the final conclusion is that "extreme care is needed to correctly interpret melt inclusions" is not new – it is well-established through studies of major element variability (Newcomb et al., 2014), hydrogen diffusive loss (Bucholz et al., 2013; Lloyd et al., 2013), and the CO₂ loss to vapor bubbles (e.g., Moore et al., 2015 and other refs presented in this manuscript).

However, the figures and the data presented are of excellent quality. The figures, specifically, present the data with appropriate error bars/statistical significance for individual data points and models, and importantly provide an amazingly large amount of data/hypothesizes in a clean, organized, well-labeled, concise, and attractive way. The figures are seemingly dense prior to reading the text, but they are actually very well-supported by the textual information.

Overall, I feel that further work might justify a resubmission although specific concerns must be addressed before a final decision is reached.

Firstly, there needs to be some systematic way to reference different eruptions, eruption locations, and eruptions periods, etc. Because such a large amount of literature data and new data, and because the HI eruptions have their own place-based jargon, it is often difficult to recall which sample/eruption the author is referring to throughout the text. Additionally, the author refers to some of the same samples in different ways, referring to dates, locations, rifts, summits, extra caldera, etc. This is problematic for a short-format journal article meant for a broader audience. Perhaps referring to them separately isn't actually necessary, and the authors can figure out a way to bin and label into 1-3 groupings and stick with a chosen definition or name for the remainder of the manuscript?

I also feel that the CO₂ story needs to be bolstered by calculated estimates of initial CO₂. There have been numerous studies that investigate the loss of CO₂ vapor bubbles, and the argument presented in this short manuscript concludes that they are unable to provide reliable barometric estimates despite a very shallow/brief comparison of the two olivine populations. I suggest the authors at least provide calculated estimates, because the CO₂-related portion of the study is one of the more novel and seemingly important in this manuscript. There is also no mention of H₂O contents and H diffusive loss, which should provide additional support for the conclusions.

Line Edits:

Line 41: "ascend beneath the summit region" is vague. Be specific about present estimate of storage from existing geochemical geophysical observations; perhaps give context of crustal thickness/MOHO depths.

Line 42: "Change rapidly with time" is vague. Be specific about "time" – days to weeks? Weeks to months? Months to years?

Line 45: "prominent cyclicity with a duration of 200 years" is vague. What kind of cyclicity? Max to max Nb/Y in 200 years? Or min to max in 200 years? Generally increasing Nb/Y or generally decreasing?

Line 83-86: After each eruption, specify within the parentheses consistently whether or not the data is from the literature. Because the "extracaldera eruption of July 1974" specifically is referred to as literature data, it seems as though all other data is not from the literature, although it must be.

Line 139: What timescale is suggested by 'prolonged' here? Can refer to timescales calculated by ref 21 for context/clarity.

Line 155: Start the sentence with "The" rather than just the trace element acronyms.

Line 159: Here, refer to the "shallow HMM" reservoir for consistency/clarity. Check use of shallow vs. deep and SC vs. HMM throughout. Why create an acronym for the reservoirs if you don't use them all the time?

Line 160: Are these the rift eruptions analyzed by this study? Or the literature? Please clarify – a first read-through of this is confusing for someone without a background in HI eruptions.

Line 163: Which 17? Confusing reference and probably unnecessary. Just say 2 of the MI populations...

Line 188: Why do the different reservoirs have to be relatively homogeneous, as mentioned in parentheses? Please explain further.

Line 201 & 209: Avoid starting a sentence with an acronym.

Line 217: Although lines 216-217 explain that CO₂ needs to be physically measured in the vapor bubble, the authors can still make some estimates of initial CO₂ without this measurement. Specifically, relevant papers by Wallace et al., 2015 and Rasmussen et al., utilize thermodynamic methodology, which only requires data that the authors have available. Although there are some discrepancies between measured and modeled corrected CO₂ values (e.g., Aster et al., 2016), the authors should be able to demonstrate differences in corrected CO₂ concentrations between melt inclusions in the two olivine populations, especially considering the difference in delta T and timescales of storage.

In addition, how do entrapment depths calculated for <Fo₈₄ olivines compare with geophysical observations of the HMM reservoir? Do they agree? Do the melt inclusions in <Fo₈₄ olivines also contain vapor bubbles? Are the vapor bubbles different sizes in the different melt inclusion populations?

Line 232: In this sentence, does entrainment refer to the same process previously referred to as scavenging? Please be consistent with this language as it is not widely accepted vocabulary, especially for a broader audience.

Line 237: This point should also be made in the CO₂ section, as it led to questions outlined above. However, it is still important to provide information about the presence or absence of vapor bubbles in this population and if so, why the depths are still well-constrained.

Line 241: This sentence is vague. Please refer to which reservoirs are being replenished, etc. That is, write this sentence to be more specific to the system at Kilauea as outlined in the text – like the SC or the HMM reservoirs that are labeled in figure 4.

Line 253: “Just prior to eruption” – what is the evidence that this occurs just prior to eruption? What is the timescale implied by “just prior?”

Line 499: Write out dates for the “Ulu period” eruptions. It is difficult for someone unfamiliar with the eruptions in HI to recall this information from the main manuscript.

Line 508: Write out the number of glass, melt inclusion, and olivine analyses performed. In particular, it is important to note the number of melt inclusions analysed from each of the 4 different samples.

Figure 1 & 2: Label/Title 4 dated eruptions from which new data was collected for this study as “Ulu period” or “this study” to better clarify the source of those data.

Reviewers' comments:

**Reviewer 1**

This well-written manuscript relating the compositions of melt inclusions, matrix glass, and
host olivines presents an interesting model for magma storage and mobilisation for the
Kīlauea system.

**We thank the review for their support of our model for Kīlauea.**

While I have some minor comments regarding the data presentation and handling, the main
issue I have is whether the findings advance our understanding of magmatic systems
conceptually.

I would like to summarise this concern by highlighting two statements in the abstract:

- (1) "Entrainment of primitive olivines into more evolved carrier melts causes crystallisation on the inclusion walls and the sequestration of CO₂ into vapour bubbles, producing spurious barometric estimates."

We address the concerns regarding the novelty of our statements regarding PEC-driven growth of bubbles in detail in the reviewers line by line comments (line 170-242 below). We have significantly expanded the discussion to emphasize that, instead of invoking PEC due to pre-eruptive cooling during fractionation, PEC occurs suddenly due to rapid thermal re-equilibration between scavenged hot, primitive olivine crystals and cooler host melts.

- (2) "Thus, the provenance of the melt inclusion record must be carefully considered before this archive is related to eruption-specific measures."

Both of these are presented as major findings and I don't see how this is new compared to what we already know. E.g. A recent article by Ruth et al. published in Nature communications shows that magma storage, mobilisation and assembly can be complex especially when considering the integration of melt inclusions and host mineral chemistry. While they don't specifically focus on mush piles in their paper, they definitely address the difficulties of 'carefully' considering the melt inclusion record. This is an example from an arc setting, however, the overall implications are the same.

We have significantly expanded the introduction section to address the known fallacies of melt inclusion records (which mostly consider the processes degrading the melt inclusion record following entrapment). We then make it clear that this manuscript addresses a more fundamental problem; even without these processes, melt inclusions in mush-rich systems may not record the pre-eruptive evolution of the magma batch of interest (Lines 28-66):

[revised manuscript text omitted]

**Finally, we emphasize that the slow rates of trace element diffusion in melt inclusions**
**means that they record crystal residence times that cannot be achieved through the**
**investigation of Fe-Mg diffusion (as conducted by Ruth et al, and numerous previous**
**studies at Kīlauea Volcano):**

*“The relatively high diffusion coefficients of these species means that profiles within*
*individual crystals only record the final processes happening decades to hours before*
*their eventual eruption^{51,52}.”*

The former statement about the need to account for CO₂ in bubbles and walls is
even more well established in multiple papers (e.g.: Aster et al. 2016, Esposito et al.
2016, Moore et al. 2015). So, I am not sure that with the exception of an interesting
mush pile model for Kīlauea what else is conceptually novel.

**We agree with the reviewers that it is well accepted in the literature that a significant**
**proportion of CO₂ in melt inclusions is found in bubbles and bubble walls. Moore et al.**
**(2015) and Aster et al. (2016) discuss how the cooling of a phenocryst after melt**
**inclusion entrapment results in the growth of a bubble due to the differential thermal**
**expansion of olivine and melt, and as a result of post entrapment crystallization.**
**Esposito et al. 2016 also demonstrate that some CO₂ is held within vapour bubbles.**

**However, few studies have investigated the processes accounting for the observation**
**that some eruptions contain large quantities of CO₂ in vapour bubbles, and others**
**relatively little. Variation in the amount of CO₂ in vapour bubbles measured by Raman**
**has been attributed to the rate of cooling upon eruption (Tucker et al., 2019). However,**
**due to rapid (and relatively similar) rates of quenching upon eruption at Kīlauea,**
**Moore et al. (2015) suggest that the CO₂ in the bubble is controlled by pre-eruptive**
**cooling following melt inclusion trapping. However, they do not expand on why this**
**would lead to different eruptions containing different proportions of CO₂ within**
**vapour bubbles.**

**In this study, we use the conceptual model revealed by melt inclusion chemistry to**
**demonstrate the relationships between mush pile processes, mixing, PEC, and CO₂**
**bubble growth at Kīlauea. Our novel conceptual proposal is that rapid thermal re-**
**equilibration between hot primitive olivine crystals scavenged from the mush pile,**
**and cooler host melts promotes extensive PEC and CO₂ sequestration into a bubble.**
**In previous studies, the bubble formation process has been linked to steady cooling**
**during fractional crystallization. (Lines 312-333):**

[revised manuscript text omitted]

Minor aspects:

It is unclear to me why new datasets were generated and supplemented with
literature data. At the end most of the key results are largely on the basis of literature
data (e.g. Figure 3b).

**New datasets were generated because previous studies measured a relatively small**
**number of inclusions per eruption (~ 10 , see table 1). With these relatively small**
**datasets, comparisons of trace element ratios in melt inclusions and matrix glasses**
**were ambiguous. Crucially, the study of Sides et al. largely focused on summit**
**eruptions, so was somewhat biased towards the more evolved crystal cargoes which**
**are phenocrysts. The ambiguity in the available literature data for extracaldera and rift**
**zone eruptions was highlighted by Tuohy et al. (2016):**

*“ The coarse nature of the olivine suggests the crystals might be cumulate in origin and*
*therefore be unrelated to the magma in which they eventually erupted (e.g., Welsch et al.,*
*2013) It is difficult, however, to know what range of Nb/Y values can distinguish*
*phenocrysts from antecrysts because Kīlauea Iki erupted mixed magmas (based on the*
*presence of diverse olivine types; Helz, 1987) that likely formed by mixing of melts with*
*some compositional variability. Such mixing can explain the compositional heterogeneity of*
*both matrix glasses and melt inclusions (Fig. 10a; see also Maclennan et al., 2003) but*
*creates ambiguity in rigorously distinguishing phenocrysts from antecrysts based on melt*
*inclusion data alone. Sides et al. (2014b) also noted a larger variation in incompatible*
*element ratios (e.g., La/Yb) for Kīlauea Iki melt inclusions compared to matrix glasses using*
*a dataset that included many more eruptive episodes. They interpreted the variability in*

*terms of mixing processes and concluded that most olivine crystals were phenocrysts*
*because the melt inclusion values bracket the matrix glasses in composition.”*

**In our study, not only do we compile all available literature data where more than 8**
**inclusions are measured per eruption, but also analyse significantly more melt**
**inclusions per eruption than previous studies (20, 27, 37, and 42, summarized in table**
**1). This combined dataset, following the subdivision of eruptions into those with**
**evolved and primitive crystal cargoes, is the first convincing demonstration that melt**
**inclusions are in trace element disequilibrium with their matrix glasses. Consideration**
**of the combined dataset (as evaluated by Sides) is far more convoluted, as the olivine**
**crystal cargoes have vastly different histories. Lastly, we specifically target a time**
**period with relatively few melt inclusion records in the literature (1969-1974). When**
**this is combined with the abundant literature data for Kīlauea Iki and Kapoho**
**eruptions (1959-1960), the broad range of matrix glass compositions means that**
**deviations between melt inclusions and glasses are more apparent.**

**We summarize the contribution of our new data in lines 119-135 and Table 1:**

*“To investigate the degree of equilibrium between erupted melts and their crystal cargoes, we*
*analysed melt inclusions and matrix glasses (for analytical details see **Methods**) from tephra*
*erupted during four eruptions temporally associated with activity at Mauna Ulu on the upper*
*East Rift Zone (ERZ) of Kīlauea (Fig. 1):*

*1) The highest fountaining phase of the Mauna Ulu eruption (December, 1969; ERZ)*

*2) The intra-caldera fissure eruption of August, 1971*

*3) The Pauahi Crater eruption (November, 1973; ERZ)*

*4) The December 1974 fissure eruption on the Seismic South West Rift Zone (SSWRZ²⁴; Fig.*
*1a-b).*

*The five-year period over which our samples were erupted includes some of the most rapid*
*historic changes in melt composition at Kīlauea (Fig. 2a). We supplement our dataset of 126*
*melt inclusions and 40 matrix glass chips with literature studies where trace elements were*
*reported in ≥ 8 inclusions, and co-erupted matrix glasses (Table 1). The combined dataset of*
*27 eruptive episodes, and 384 melt inclusions spans ~ 600 years of eruptive history at Kīlauea*
*and incorporates large variations in matrix glass (Fig. 2b-c) and whole rock compositions (Fig.*
*2a)³⁵. “*

*...At the end most of the key results are largely on the basis of literature data (e.g. Figure*
*3b).*

**While the figure regarding CO₂ contents relies on the data collected by Sides et al.,**
**the trends we describe rely on the subdivision of eruptions into primitive and evolved**
**crystal cargoes that we have developed in this study. This classification scheme**
**relied on the extensive compilation of olivine forsterite data conducted in this study,**
**and the conceptual model we develop of olivine-host relationships.**

*I am also wondering about what the reference millenia-style storage is based on (fig. 2 and*
*text references). The vast majority of the variations in Nb/Y is accommodated in the last*
*<100 years (Fig 2a). So, the entire dynamics could still be reflecting that timeframe not*
*requiring a long millenia scale mush (while I personally could see it existed, the data is not*
*showing that).*

**We thank the reviewer for pointing out the lack of clarity in this section. Although**
**$\sim 43\%$ of the Nb/Y variation observed in the Dec 1969 melt inclusion population has**
**been observed in whole-rock compositions between 1934 and 1982 (the downgoing**
**limb on figure 2a), melt inclusion populations must have been trapped before the**

eruption date. Thus the presence of Nb/Y ratios as low as 0.37 within melt inclusions
from the 1959 eruption (and similarly low for 1969-1974) indicates that melts were
trapped before Nb/Y ratios started increasing at ~1800 AD (requiring storage for >170
326 years). The large number of melt inclusions with Nb/Y higher than the maximum
observed in whole-rock compositions may imply that storage times significantly
exceed a few centuries (extending back to the last time that such enriched melts were
available within the plumbing system). We have clarified these arguments in the text
in the new section of the discussion entitled “Centurial storage times within mush
piles”, Lines 268-279, and by marking on the 1959 eruption on Fig. 2”.

*“An estimate of the minimum residence times of primitive olivine crystals in mush piles can be
obtained by comparing the range of trace element ratios in melt inclusions to erupted lava
compositions. Bulk-rock and glass analyses from the 1959 Kīlauea Iki eruption lie close to
upper limit of Nb/Y ratios observed since ~1790 (Fig. 2a). Yet, the vast majority of co-erupted
melt inclusions have significantly lower Nb/Y ratios (Fig. 2c), down to ~0.4. Melts with these
compositions were only present in Kīlauea’s plumbing system before ~1790 AD (Fig. 2a),
suggesting that these erupted crystals were stored for at least 170 years. It is highly possible
that crystals were stored for considerably longer timescales; the range of Nb/Y ratios in melt
inclusion records from the three rift eruptions investigated in this study greatly exceed the
range of bulk rock compositions between 1790-1982. In fact, the 1969 eruption displays trace
element diversity surpassing the range of erupted lava compositions over a 350 kyr period
(Fig. 2c).”*

If you think the changes in CO₂ are related to post-entrapment crystallisation and that Nb/Y
is not affected by it, while incompatible element concentrations are getting enriched, it would
be interesting to see this relationship between Nb and PEC or Y and PEC.

**We have added figures into the supplementary information (Fig. E and F) showing that**
**there is no statistically significant correlation between the amount of PEC or MgO,**
**and the Nb and Y concentrations, or the Nb/Y ratio (pasted below)**

How important is the high Nb/Y glass matrix sample to account for the excellent R² in figure
2b. While that relationship is still there without this subset of data, it seems to be heavily
weighted towards it.

**The high Nb/Y sample is from 1961 – While this sample definitely contributes to the**
**excellent R² and p values shown in the main text, removal of this sample still**
**produces a very strong correlation. We mention this in the figure caption (Lines 481-**
**483):**

*“While the 1961 summit eruption (Nb/Y_{Glass} ~0.89) certainly strengthens the observed*
*correlation in b), the regression is still very good if this eruption is excluded (R²=0.84, p=10⁻⁴).*”

How much do you have to worry about diffusion effects on Nb and Y in the melt inclusions
through diffusion in the olivine? While Cottrell et al (2002) suggested that incompatible
elements should be safe, they also stated: “Examination of published trace element
concentrations of olivine hosted inclusions show little evidence for re-equilibration, at least for
the light REE and other highly incompatible elements. It is difficult, however, to provide firm
constraints due to the uncertainties in olivine diffusivities and the initial condition.”
Diffusivities of trace elements in olivine has been a major debate in the last few years
starting with Spandler and O’Neill (2010). If diffusivities in olivine are much greater than
initially thought (especially for high Si activity systems) then fidelity of the trace element
record in the melt inclusions may also be compromised, especially for millennia-old mush
pile melt inclusions.

**We have added a discussion about trace element diffusion in the section “centurial**
**storage times within mush piles”. More recent papers such as Cherniak (2010) show**
**that diffusivities of REE are 3 orders of magnitude slower than those suggested by**
**Spandler et al. 2007. Furthermore, it has recently been shown using spinel diffusion**
**profiles that olivines erupted in the Icelandic Borgarhaun flow are stored for a mean**
**time of ~1400 years (Mutch et al., 2019; Science). Yet, these melt inclusions display**
**considerable trace element heterogeneity that correlates strongly with forsterite. Such**
**relationships would be erased by trace element re-equilibration. Finally, our dataset**
**shows similar correlations between highly incompatible trace elements (variability**
**produced by varying melt extents in the mantle) for glasses and melt inclusions.**
**Diffusive re-equilibration of these elements within melt inclusions would destroy**
**these correlations (Lines 292-310).**

*“A related question is whether trace elements within olivine-hosted melt inclusions undergo*
*diffusive equilibration with their surrounding melts during centurial storage. While early*
*estimates of diffusion rates for rare earth elements suggest that diffusive re-equilibration may*
*occur over tens to hundreds of years⁵⁴, more recent studies calculate diffusivities that are ~3*
*orders of magnitude lower (requiring $10^4 - 10^6$ years for 50% re-equilibration of Ce and Yb in*
*a 50 μm melt inclusion within a ~1 mm olivine)⁵⁵. Evidence for the lack of trace element re-*
*equilibration in our dataset is provided by the similarity of regression lines for incompatible*
*elements defined by matrix glasses and melt inclusions (e.g. Nb vs. La; Supplementary Fig*
*C). These strong correlations are likely produced by different extents of mantle melting³³. If*
*trace element re-equilibration was occurring within melt inclusion populations during*
*prolonged mush pile storage, different extents of re-equilibration for different trace elements*
*(and different inclusion and host olivine sizes) would result in melt inclusions defining more*
*scattered correlations with different regression lines compared with matrix glasses. While*
*there are no available experimental estimates for the diffusivities of Nb and Y in olivine, the*

*similar correlations defined by melt inclusions and matrix glasses for Nb vs. La*
*(Supplementary Fig. C) and Y and Yb (Supplementary Fig. C) suggests that these elements*
*are also resistant to diffusive re-equilibration during centurial storage. Thus, comparisons of*
*melt inclusion diversity to erupted melt compositions provides the most reliable estimate of*
*storage timescales in long-lived mush-dominated systems. “*

**References:**

- • Moore LR, Gazel E, Tuohy R, Lloyd AS, Esposito R, Steele-MacInnis M, Hauri EH, Wallace
PJ, Plank T, Bodnar RJ. Bubbles matter: An assessment of the contribution of vapor bubbles
to melt inclusion volatile budgets. *American Mineralogist*. 2015 Apr 1;100(4):806-23.
• Esposito R, Lamadrid HM, Redi D, Steele-MacInnis M, Bodnar RJ, Manning CE, De Vivo
B, Cannatelli C, Lima A. Detection of liquid H₂O in vapor bubbles in reheated melt
inclusions: Implications for magmatic fluid composition and volatile budgets of magmas?.
*American Mineralogist*. 2016 Jul 1;101(7):1691-5.
• Aster EM, Wallace PJ, Moore LR, Watkins J, Gazel E, Bodnar RJ. Reconstructing CO₂
concentrations in basaltic melt inclusions using Raman analysis of vapor bubbles. *Journal of*
*Volcanology and Geothermal Research*. 2016 Sep 1;323:148-62.
• Ruth, D.C., Costa, F., de Maisonrouve, C.B., Franco, L., Cortés, J.A. and Calder, E.S.,
2018. Crystal and melt inclusion timescales reveal the evolution of magma migration before
eruption. *Nature communications*, 9(1), p.2657.
• Cottrell E, Spiegelman M, Langmuir CH. Consequences of diffusive reequilibration for the
interpretation of melt inclusions. *Geochemistry, Geophysics, Geosystems*. 2002 Apr
1;3(4):1-26.
• Spandler C, O'Neill HS. Diffusion and partition coefficients of minor and trace elements in
San Carlos olivine at 1,300 C with some geochemical implications. *Contributions to*
*Mineralogy and Petrology*. 2010 Jun 1;159(6):791-818.

**Reviewer 2**

This paper presents an interesting and novel study of the relationship between melt
inclusion bearing crystals and their carrier melts at Kīlauea volcano. It provides important
context for studies using melt inclusions (from Hawaii and elsewhere) and shows in a simple
but convincing way that the inclusions in the more primitive crystals are probably the least
faithful recorders of pre-eruptive conditions. This is contrary to what is generally assumed. It
is well written and the data looks to be of high quality. I have no major concerns that would
be a barrier to publication. The data presented certainly supports the conclusions.

**We thank the reviewer for their support of our work, particularly regarding the**
**implications of this study for the interpretation of melt inclusions worldwide. We have**
**emphasized the point from the reviewer that primitive melt inclusions provide the**
**most unreliable record in the conclusion (Lines 408-410):**

*“Overall, these findings challenge the common assumption that the most primitive crystals*
*provide the most pristine record of pre-eruptive processing.”*

Prior to publication, however, they may want to consider adding some additional discussion
on some of the unexplored aspects of their data/model. For example, why do the higher Fo
olivines trap (on-average) more enriched melts than the lower Fo crystals?

**We believe that the fact that the more evolved olivines tend to host melt inclusions**
**with lower Nb/Y ratios is a result of their eruption date, rather than any magmatic**
**process. The vast majority of the rift eruptions investigated in this study occur while**
**glass Nb/Y ratios were high (Fig. 2a), and their melt inclusions trap the range of Nb/Y**
**ratios present in the plumbing system over several centuries. In contrast, with the**
**exception of the 1961 summit eruption, the intracaldera summit eruptions occurred**
**when melt Nb/Y ratios were low, so these phenocrystic crystal cargoes also have low**
**Nb/Y ratios. We have not addressed this point in the main text, as we do not feel it**
**provides any insight into Kīlauea’s plumbing system.**

Is the implication that all MIs that have compositions that differ from the carrier melt are
antecrysts? Or do they see a role for some of the MIs in recording the melt mixing process?

**We agree that the mixing will create some trace element variability in melt inclusions**
**records. A minimum estimate of the variability generated by magma mixing is**
**provided by the range of trace element ratios in melt inclusions from the 1971 summit**
**eruption (Fig. 2b; range of Nb/Y~0.14). The range of Nb/Y ratios in the 1969 eruption is**
**~0.57, thus mixing may contribute ~25% of the observed variation.**

**We have discussed this in the text in lines 258-266:**

*“An approximate estimate of the contribution to the diversity of melt inclusions from reservoir*
*heterogeneity and magma mixing can be obtained from the range of trace element ratios in*
*melt inclusions from the 1971 summit eruption (Fig. 2b; Nb/Y=0.6-0.74). This accounts for*
*only 25% of the variation in Nb/Y ratios observed in the 1969 eruption (Fig. 2c; Nb/Y=0.37-*
*0.94). Thus, indistinguishable melt inclusion populations with a broad range of melt inclusion*
*trace element ratios in many different primitive eruptions, combined with the lack of*
*relationship with forsterite contents, supports a model in which carrier melts randomly*
*scavenge olivine antecrysts from mush piles containing highly diverse melt inclusion*
*populations just prior to eruption²”*

While the mean of the lower Fo MIs matches that of the carrier melt, the range of values
from several of the eruptions covers most of the 350 kyr range in Nb/Y. What do these
outlier values represent?

**The main outlier on Fig. 2 b (glass Nb/Y>0.8) represents data from Sides et al. for the**
**1961 summit eruption. Detailed examination of Nb/Y systematics Vs. Fo reveal that**
**three inclusions have significantly lower ratios than co-erupted matrix glasses. As we**
**do not possess these samples for further examination, it is difficult to know the origin**
**of these outliers in literature data.**

**Reviewer 3**

**Key Results/Manuscript Summary:**

The manuscript titled, "Crystal scavenging from mush piles recorded by melt inclusions,"
provides an evaluation of new and literature-derived in-situ geochemical analysis of glasses,
melt inclusions, and olivines from the tephra and lava of historic eruptions from Kīlauea
Volcano in Hawaii. The data analysis of magmas from numerous eruptions identifies two
main olivine populations: a lower Fo olivine population which hosts melt inclusions derived
from a relatively shallow, magmatic reservoir with melt inclusions related to the matrix
glasses and a higher Fo populations (more primitive) that hosts compositional
heterogeneous melt inclusions populations, which are likely stored in deeper reservoirs
(despite barometric estimates from CO₂/H₂O, which the authors suggest are
underestimated because of post-entrapment processes), that are not in equilibrium with their
host glasses. These results are used to hypothesize a model for a storage and ascent of
Hawaiian magmas that involves scavenging of crystal cargo from more deeply stored
reservoirs with heterogeneous melt inclusion populations. The manuscript warns that the
presence of 'antecrysts' is a complexity that is imperative to identify in any studies utilizing
melt inclusions.

**General Comments:**

While this manuscript provides an interesting case study and uniquely combines a variety of
recently established hypotheses in the field of melt inclusion geochemistry, I do not feel that
the manuscript adheres to the four main criteria for publication in a Nature Research journal.
The manuscript is well-written and the data and analysis provide strong support for the
presented conclusions, but it does not strike me to be "of extreme importance to scientists"
in my field likely not "interesting to researchers in other related disciplines." This is actually
highlighted by the manuscript itself which relies heavily on previous interpretations to provide
a framework for their study. For example, evidence for 2 separate storage reservoirs at
Kīlauea was first suggested based on Pb isotope distinctions (ref 15), and forsterite
composition peaks previously attributed to crystal mush pile processes in Iceland (ref 21).
Additionally, the final conclusion is that "extreme care is needed to correctly interpret melt
inclusions" is not new – it is well-established through studies of major element variability
(Newcomb et al., 2014), hydrogen diffusive loss (Bucholz et al., 2013; Lloyd et al., 2013),
and the CO₂ loss to vapor bubbles (e.g., Moore et al., 2015 and other refs presented in this
manuscript).

**We have refocused the manuscript to address these novelty concerns. Firstly, we**
**have added significant detail into the discussion regarding the known fallacies of the**
**melt inclusion record (Lines 28-66):**

*“However, it is becoming increasingly apparent that melt inclusions are not a perfect archive*
*of magmatic processes occurring at depth. The post-entrapment crystallization (PEC) of*
*olivine on the walls of the melt inclusion during cooling of the host crystal with progressive*
*fractional crystallization, or upon eruption, changes the major and trace element composition*
*of the remaining melt^{5,6}. The concentration of elements which are compatible in olivine*
*decrease (e.g., MgO, Ni), while incompatible elements increase (e.g. Nb, La, Sm, H₂O)⁷.*
*These changes, combined with a drop in inclusion pressure, favour the formation of a CO₂-*
*rich vapour bubble⁶⁻⁸. Unless the CO₂ content of the bubble is quantified (e.g., using Raman*
*spectroscopy), melt inclusion analyses by techniques such as secondary ion mass*
*spectrometry (SIMS) or Fourier Transform Infra Red spectroscopy (FTIR) will underestimate*
*the CO₂ content at the time of entrapment⁸⁻¹⁰. Furthermore, global compilations of melt*
*inclusions demonstrate that the process of decrepitation, where the host olivine ruptures and*
*releases CO₂ due to a large pressure difference between the inclusion and the host melt,*
*accounts for the significantly lower entrapment pressures recorded by melt inclusions than*
*independent petrological barometers (e.g., clinopyroxene-liquid)⁷.*

*Rapid diffusion rates of H⁺ in olivine mean that melt inclusion water contents are also*
*vulnerable to diffusional re-equilibration¹¹. This process may produce anomalously low water*
*contents if the sample is not quenched rapidly upon cooling (allowing the melt inclusion to*
*equilibrate with the degassed carrier melt)^{12,13}, or anomalously high water contents due to*
*entrainment into a water-rich carrier melt, or the mixing of compositionally diverse melts¹⁴.*
*Finally, a recent study at Llaima Volcano combining melt inclusion volatile data with diffusive*
*modelling of major element zoning in host olivines demonstrated that melt inclusions record*
*the progressive mixing of melts stored at various levels in the plumbing system for months to*
*years prior to their eventual eruption¹⁵.*

*However, in addition to the processes discussed above which alter melt inclusion*
*geochemistry pre- and post-entrapment, the increasingly prevalent view of magmatic systems*
*as mush-dominated¹⁶ raises more fundamental questions regarding the utility of melt*
*inclusions. Settled crystals may be stored at a wide range of depths within extensive cumulate*
*piles within the crust for many millenia¹⁶⁻¹⁸. The re-entrainment of these crystals into unrelated*
*magma batches challenges the common assumption that melt inclusions and matrix glasses*
*(the solidified ‘carrier’ melt) are related¹⁹, such that inclusions provide a record of the pre-*
*eruptive storage and evolution of the erupted melt. Instead, a significant proportion of erupted*
*crystals may be “antecrysts”; commonly defined as crystals which formed in a separate*
*magma batch to the one in which they were erupted^{1,17,20}. Here we assess crystal-melt*
*relationships using olivine-hosted melt inclusions from Kīlauea Volcano, Hawai‘i, to assess*
*the utility of melt inclusion records in a mush-dominated volcanic system “*

**We then emphasize that our findings that melt inclusions hosted in primitive olivines**
**are genetically unrelated to the carrier melts is a more fundamental issue with melt**
**inclusion records. Even if secondary processes such as H diffusion, PEC, and bubble**
**growth occur, these melt inclusions still would not inform us about the pre-eruptive**
**storage of their parental melt (lines 427-431).**

*“While the processes acting to degrade the melt inclusion record following entrapment are*
*well recognised, we demonstrate that the common assumption that melt inclusions record*
*pre-eruptive processes is flawed in mush-rich systems. However, scavenged melt inclusion*
*records are far from redundant; they provide novel insights into the storage, and subsequent*
*remobilization, of crystals in magmatic mush piles.”*

**Our study is the first to assess melt inclusion records in mush-rich systems – our**
**findings have implications for a wide variety of high melt flux systems that are likely**
**characterized by extensive cumulate piles (Lines 423-427):**

*“The history of pre-eruptive processes preserved in melt inclusions can be obscured*
*by accumulation and storage of inclusion-bearing crystals in mushes, followed by*
*crystal scavenging by new carrier liquids. This decoupling of inclusions and crystals*
*from their carrier liquids may be commonplace at volcanoes where high melt fluxes*
*create long-lasting magma reservoirs and associated mush piles”*

**We also emphasize in the revised manuscript that the centurial storage times we**
**estimate from trace element diversity in melt inclusions are unprecedented at Kīlauea**
**(and many other volcanoes), as most studies focus on the diffusion of Fe-Mg (Lines**
**280-284):**

*“Our study is the first to recognise centurial storage timescales of crystal mushes at*
*Kīlauea. Most previous estimates of timescales are based on the interdiffusion of Fe*
*and Mg within olivine^{51,52}. The relatively high diffusion coefficients of these species*
*means that profiles within individual crystals only record the final processes*
*happening decades to hours before their eventual eruption^{51,52}.”*

However, the figures and the data presented are of excellent quality. The figures,
specifically, present the data with appropriate error bars/statistical significance for individual
data points and models, and importantly provide an amazingly large amount of
data/hypothesizes in a clean, organized, well-labeled, concise, and attractive way. The
figures are seemingly dense prior to reading the text, but they are actually very well-
supported by the textual information.

**We thank the reviewer for their support of our data presentation.**

Overall, I feel that further work might justify a resubmission although specific concerns must
be addressed before a final decision is reached.

**We hope that the reviewer finds the new focus of the manuscript on the novelty of**
**determining crystal residence times from melt inclusions, and the wider implications**
**of our study for the interpretation of melt inclusion records in mush-rich systems**
**acceptable.**

Firstly, there needs to be some systematic way to reference different eruptions, eruption
locations, and eruption periods, etc. Because such a large amount of literature data and
new data, and because the HI eruptions have their own place-based jargon, it is often
difficult to recall which sample/eruption the author is referring to throughout the text.
Additionally, the author refers to some of the same samples in different ways, referring to
dates, locations, rifts, summits, extra caldera, etc. This is problematic for a short-format
journal article meant for a broader audience. Perhaps referring to them separately isn't
actually necessary, and the authors can figure out a way to bin and label into 1-3 groupings
and stick with a chosen definition or name for the remainder of the manuscript?

**We thank the reviewer for pointing out that the eruption-specific detail was difficult to**
**follow. We have now added a table into the text, which allows readers to identify the**
**eruption by year (important information for people who work on Kīlauea specifically),**
**the location (intracaldera, extracaldera, rift zone) and the classification based on**
**forsterite content.**

	Date	Mean Fo content	Location	Reference	# of MI
$\overline{Fo} > Fo_{84}$	1832	86.0	Extracaldera	Sides et al.	9
	1959 (Ep 1, 2, 3, 5, 6, 7, 8, 15, 16)	87.5, 86.8, 86.1, 87.2, 87.1, 85.9, 86.8, 86.4, 86.0	Extracaldera	Sides et al.	10, 11, 13, 8, 9, 12, 10, 10, 15
	1960	85.7	ERZ	Sides et al.	17
	1960 (Kap6, Kap8)	85.5, 88.4	ERZ	Tuohy et al.	10, 19
	Dec 1969	86.7	ERZ	This study	37
	Nov, 1973	85.6	ERZ	This study	27
	July, 1974	85.4	Extracaldera	Sides et al.	8
	Dec, 1974	87.3	SWRZ	This study	42
$\overline{Fo} < Fo_{84}$	1445	80.6	Intracaldera	Sides et al.	12
	1500	83.3	Intracaldera	Sides et al.	9
	1885	81.5	Intracaldera	Sides et al.	10
	1961	82.4	Intracaldera	Sides et al.	9
	Aug, 1971	82.8	Intracaldera	This study	20
	1982	82.2	Intracaldera	Sides et al.	9
	2008 (3 episodes)	82.6, 82.6, 82.9	Intracaldera	Sides et al.	9, 20, 9
	2010	81.7	Intracaldera	Sides et al.	10

I also feel that the CO₂ story needs to be bolstered by calculated estimates of initial CO₂.
There have been numerous studies that investigate the loss of CO₂ vapor bubbles, and the
argument presented in this short manuscript concludes that they are unable to provide
reliable barometric estimates despite a very shallow/brief comparison of the two olivine
populations. I suggest the authors at least provide calculated estimates, because the CO₂-
related portion of the study is one of the more novel and seemingly important in this
manuscript.

**We acknowledge that there are several studies in the literature that estimate initial**
**CO₂ contents from measured bubble sizes and the CO₂ equation of state (e.g., Tucker**
**et al. 2019). However, our own Raman measurements on bubbles from multiple**
**different Kīlauean eruptions (in prep) reveal that a large proportion of vapour bubbles**
**contain quantities of CO₂ below detection limit (particularly in olivines which have**
**experienced limited PEC). This is likely driven by the continued expansion of the**
**vapour bubble above the glass transition temperature, but temperature-limited**
**diffusion of CO₂ into the growing vapour bubble (Anderson and Brown, 1993; Wallace**
**et al. 2015; Maclennan, 2017). Thus, the observed size of bubbles in melt inclusions**
**which have undergone extensive PEC reflects a combination of expansion upon**
**eruption, and bubble formation during crystal scavenging. Without measuring the**
**CO₂ content of these melt inclusions by Raman (impossible in literature data; even if**
**the samples were obtained, the bubbles have already been polished through), it is not**
**possible to estimate the total CO₂ content. We address these points in lines 359-364:**

*“Thus, melt inclusion CO₂ contents in evolved crystal cargoes produce reliable barometric*
*estimates, even though these inclusions contain bubbles. This implies that these bubbles*
*only contain a small fraction of the total CO₂ budget. CO₂-poor bubbles may form during*
*post-eruptive cooling, due to differences between the glass transition temperature and the*
*temperature at which C-diffusion becomes extremely slow (allowing bubble grow, but*
*hindering the diffusion of CO₂ from the melt into the bubble)^{7,60,61}”.*

There is also no mention of H₂O contents and H diffusive loss, which should provide
additional support for the conclusions.

**We have added a section on H⁺ loss into the introduction (Lines 45-49):**

*“Rapid diffusion rates of H⁺ in olivine mean that melt inclusion water contents are*
*also vulnerable to diffusional re-equilibration¹¹. This process may produce*
*anomalously low water contents if the sample is not quenched rapidly upon cooling*
*(allowing the melt inclusion to equilibrate with the degassed carrier melt)^{12,13}, or*
*anomalously high water contents due to entrainment into a water-rich carrier melt, or*
*the mixing of compositionally diverse melts¹⁴.”*

**It is hard to assess the reliability of the H⁺ record in literature data as we do not have**
**access to information required for quantitative modelling such as crystal size, and**
**distance of the inclusion from the edge of the crystal. it is plausible that H⁺ is reset**
**during transport to match that of the carrier melt. However, this is hard to deconvolve**
**from loss of H⁺ upon eruption, as the literature samples were variably quenched**
**(some are spatter, some reticulite, some small lava flows).**

Line Edits:

Line 41: “ascend beneath the summit region” is vague. Be specific about present estimate of
storage from existing geochemical geophysical observations; perhaps give context of crustal
thickness/MOHO depths.

**We have added more detail into this section (Lines 71-80):**

*“Primitive basaltic magmas supplied from the Hawaiian hotspot at > 100 km depth²¹*
*ascend through the lithosphere into two main crustal storage reservoirs situated*
*beneath the summit of Kīlauea^{22–24}. Geophysical observations indicate that the*
*deeper, South Caldera (SC) reservoir is located at ~2-6 km depth^{22,25}, while the*
*shallower Halema‘ūma‘u (HMM) reservoir is located at ~1 km depth²⁴. The presence*
*of two distinct mixing trends in Pb isotope ratios of lavas erupted since the 1970s*
*corroborates geophysical evidence that magma is stored in two main reservoirs^{26,27}.*
*A combination of geophysical and geochemical observations suggests that the SC*
*reservoir supplies magma to extra-caldera and rift zone eruptions^{26,27}, while the*
*HMM reservoir supplies intra-caldera summit eruptions and summit lava lakes^{22,25}.”*

Line 42: “Change rapidly with time” is vague. Be specific about “time” – days to weeks?
Weeks to months? Months to years?

**We have clarified this sentence by adding specific examples of the geochemical**
**variations (Lines 95-100):**

*“Ratios of elements with similar incompatibility during crystal fractionation (e.g. Nb/Y, La/Yb)*
*and isotopic ratios (e.g. ²⁰⁶Pb/²⁰⁴Pb, ⁸⁷Sr/⁸⁶Sr) show pronounced changes over decadal to*
*centurial timescales, resulting from heterogeneity in the mantle source³², conditions of*
*melting³³, and incomplete melt mixing during magma ascent and storage^{34,26,27}. For example,*
*Nb/Y increases from ~0.4 to 0.7 between ~1800 AD and 1930 AD, before falling again to*
*~0.49 in 1982³⁵. Concurrently, ²⁰⁶Pb/²⁰⁴Pb rises from ~18.40 to ~18.65 and back to*
*~18.40²⁶.”*

Line 45: “prominent cyclicity with a duration of 200 years” is vague. What kind of cyclicity?

Max to max Nb/Y in 200 years? Or min to max in 200 years? Generally increasing Nb/Y or
generally decreasing?

**As above, we have clarified this section regarding geochemical cyclicity by giving**
**specific examples of the variations shown for Nb/Y and Pb/Pb isotopes (Lines 98-100):**

*“For example, Nb/Y increases from ~0.4 to 0.7 between ~1800 AD and 1930 AD, before*
*falling again to ~0.49 in 1982³⁵. Concurrently, ²⁰⁶Pb/²⁰⁴Pb rises from ~18.40 to ~18.65 and*
*back to ~18.40²⁶.”*

Line 83-86: After each eruption, specify within the parentheses consistently whether or not
the data is from the literature. Because the “extracaldera eruption of July 1974” specifically is
referred to as literature data, it seems as though all other data is not from the literature,
although it must be.

**We thank the reviewer for pointing out the ambiguity in this sentence, we have which**
**eruptions we analyse, and which are from the literature (Lines 145-154):**

*“Primitive crystal cargoes were observed in the three rift eruptions analysed in this*
*study (1969, 1973, 1974; red, blue and black diamonds in Fig. 3a), and in 14 eruptive*
*episodes from the literature (1832 and July 1974 eruption, 9 episodes of the 1959*
*Kīlauea Iki eruption, and 3 episodes of the 1960 Kapoho eruption; magenta diamonds*
*in Fig. 3a)^{3,19}. Evolved olivine compositions are observed in the intra-caldera eruption*
*of 1971 (this study; green triangles in Fig. 3a), and 9 eruptive episodes from the*
*literature (1500, 1885, 1961, and 1982 eruptions, 3 episodes of the 2008 summit*
*eruption, and 2 episodes of the 2010 summit eruptions; cyan triangles in Fig. 3a)³.*
*Primitive crystal cargoes are significantly out of major element equilibrium with their*
*matrix glasses, while evolved crystal cargoes lie close to the equilibrium composition*
*(Fig. 3a).”*

**We have also added a table (Table 1) to address reviewer comments that it is hard to**
**follow the different references to eruptions (date, location, and study they were**
**analysed in). We hope this will allow readers who are not familiar with Kīlauea to**
**follow the various classification schemes used (e.g. mean forsterite content, location,**
**etc.)**

	Date	Mean Fo content	Location	Reference	# of MI
FO > FO ₈₄	1832	86.0	Extracaldera	Sides et al.	9
	1959 (Ep 1, 2, 3, 5, 6, 7, 8, 15, 16)	87.5, 86.8, 86.1, 87.2, 87.1, 85.9, 86.8, 86.4, 86.0	Extracaldera	Sides et al.	10, 11, 13, 8, 9, 12, 10, 10, 15
	1960	85.7	ERZ	Sides et al.	17
	1960 (Kap6, Kap8)	85.5, 88.4	ERZ	Tuohy et al.	10, 19
	Dec 1969	86.7	ERZ	This study	37
	Nov, 1973	85.6	ERZ	This study	27
	July, 1974	85.4	Extracaldera	Sides et al.	8
	Dec, 1974	87.3	SWRZ	This study	42
FO < FO ₈₄	1445	80.6	Intracaldera	Sides et al.	12
	1500	83.3	Intracaldera	Sides et al.	9
	1885	81.5	Intracaldera	Sides et al.	10
	1961	82.4	Intracaldera	Sides et al.	9
	Aug, 1971	82.8	Intracaldera	This study	20
	1982	82.2	Intracaldera	Sides et al.	9
	2008 (3 episodes)	82.6, 82.6, 82.9	Intracaldera	Sides et al.	9, 20, 9
	2010	81.7	Intracaldera	Sides et al.	10

Line 139: What timescale is suggested by 'prolonged' here? Can refer to timescales
calculated by ref 21 for context/clarity.

**We have added a discussion into the text of the ambiguity of timescales from**
**forsterite diffusion in mush piles within an open system (Kilauea) compared to closed**
**system fractionation in a sill (used by Thomson and MacLennan in Iceland) in Lines**
**200-208:**

*“Assessing storage timescales from peaked forsterite distributions requires knowledge of the*
*initial distribution, and the mush pile height. Thomson and MacLennan³⁸ explored this*
*parameter space for Icelandic lavas assuming that a sill of variable thickness underwent*
*progressive fractional crystallization in a closed system. However, the SC reservoir is an open*
*system, with primitive melts entering at the base, and variably evolved melts leaving the*
*reservoir to be erupted, or stored within the rift zones⁴². Repeated injection of more primitive*
*melts likely produces cyclic variations in forsterite content with height. Clearly, an alternative*
*approach is required to assess the residence times of primitive olivine crystals in open (and*
*therefore highly unconstrained) systems such as Kilauea.“*

Line 155: Start the sentence with “The” rather than just the trace element acronyms.

**Amended**

Line 159: Here, refer to the “shallow HMM” reservoir for consistency/clarity. Check use of
shallow vs. deep and SC vs. HMM throughout. Why create an acronym for the reservoirs if
you don't use them all the time?

**We have amended the use of these acronyms for consistency.**

Line 160: Are these the rift eruptions analyzed by this study? Or the literature? Please clarify

– a first read-through of this is confusing for someone without a background in HI eruptions.

We have amended this sentence to clarify that these rift eruptions were analysed in this study (Lines 227-228):

“In contrast, melt inclusions from the three rift eruptions investigated in this study (1969, 1973, 1974) ..”

Line 163: Which 17? Confusing reference and probably unnecessary. Just say 2 of the MI populations...

We feel it is important to emphasize that we have investigated 17 different eruptions, and only 2 of these 17 show statistically significant differences. We have added a reference to Table 1 to clarify which eruptions we are discussing (230-231):

“In fact, of the 17 eruptions with primitive crystal cargoes (Table 1), only two of the melt inclusion populations have distinguishable means at $\alpha=0.05$.”

Line 188: Why do the different reservoirs have to be relatively homogeneous, as mentioned in parentheses? Please explain further.

We have clarified these reasons in the preceding paragraph, and in this paragraph: Lines 243-245:

“In contrast, Kīlauean melt inclusions exhibit no obvious correlation between trace element diversity and olivine forsterite contents (Supplementary Fig. B). Additionally, the presence of remarkably coherent temporal variations in lava geochemistry at widely-spaced eruption sites at Kīlauea suggests that erupted lava compositions represent the composition of a well-mixed reservoir^{27,41}”

Lines 254-258:

“Regardless of the exact mechanism producing the relatively homogenous reservoir compositions, the apparent absence of diverse melt compositions within the plumbing system based on erupted lava compositions implies that diverse melt inclusion populations were acquired from many different, well-mixed reservoir compositions present in the plumbing system over prolonged periods..”

Line 201 & 209: Avoid starting a sentence with an acronym.

We have rephrased these sentences.

Line 217: Although lines 216-217 explain that CO₂ needs to be physically measured in the vapor bubble, the authors can still make some estimates of initial CO₂ without this measurement. Specifically, relevant papers by Wallace et al., 2015 and Rasmussen et al., utilize thermodynamic methodology, which only requires data that the authors have available. Although there are some discrepancies between measured and modeled corrected CO₂ values (e.g., Aster et al., 2016), the authors should be able to demonstrate differences in corrected CO₂ concentrations between melt inclusions in the two olivine populations, especially considering the difference in delta T and timescales of storage.

As discussed in lines 661-681 of this rebuttal document, we are reluctant to estimate CO₂ using the equation of state methods. Our reluctance stems from our observations in relatively evolved Kīlauean eruptions that many vapour bubbles have CO₂ contents below the detection limit of Raman Spectroscopy. We emphasize issues

**with measured bubble volumes (which may continue to grow after CO₂ diffusion**
**halts) in lines 359-364:**

*“Thus, melt inclusion CO₂ contents in evolved crystal cargoes produce reliable barometric*
*estimates, even though these inclusions contain bubbles. This implies that these bubbles*
*only contain a small fraction of the total CO₂ budget. CO₂-poor bubbles may form during*
*post-eruptive cooling, due to differences between the glass transition temperature and the*
*temperature at which C-diffusion becomes extremely slow (allowing bubble growth, but*
*hindering the diffusion of CO₂ from the melt into the bubble)^{7,60,61}.*

In addition, how do entrapment depths calculated for <Fo84 olivines compare with
geophysical observations of the HMM reservoir? Do they agree?

**We have emphasized in the text that these pressures overlap in lines 355-360:**

*“In contrast, the concentration of CO₂ in melt inclusions hosted within evolved crystal*
*cargoes indicate entrapment pressures between ~8-75 MPa³, with most inclusions clustering*
*between 25-50 MPa³. These pressures encompass geophysical constraints on the depth of*
*the HMM reservoir (pressures of ~25-50 MPa for storage depths of 1-2km²⁴ and densities of*
*~2600 kg/m³⁴²). Thus, melt inclusion CO₂ contents in evolved crystal cargoes produce*
*reliable barometric estimates, even though these inclusions contain bubbles”*

Do the melt inclusions in <Fo84 olivines also contain vapor bubbles? Are the vapor bubbles
different sizes in the different melt inclusion populations?

**We observe vapour bubbles in the melt inclusions from the 1971 summit eruption,**
**and there are no obvious differences in bubble sizes between these samples and the**
**rift eruptions. However, we do not have CO₂ data for these samples. As discussed in**
**Tucker et al. 2019, it is very difficult to determine the true size (or occurrence) of**
**vapour bubbles in samples which have already been ground down (as is the case for**
**the small number of samples remaining in Cambridge from the Sides et al. study). We**
**have addressed this in the text in lines 360-364:**

*“This implies that these bubbles only contain a small fraction of the total CO₂ budget. CO₂-*
*poor bubbles may form during post-eruptive cooling, due to differences between the glass*
*transition temperature and the temperature at which C-diffusion becomes extremely slow*
*(allowing bubble grow, but hindering the diffusion of CO₂ from the melt into the bubble)^{7,60,61}.”*

Line 232: In this sentence, does entrainment refer to the same process previously referred to
as scavenging? Please be consistent with this language as it is not widely accepted
vocabulary, especially for a broader audience.

**We thank the reviewer for pointing out this inconsistent terminology, we have now**
**used “scavenged” at every relevant point in the text.**

Line 237: This point should also be made in the CO₂ section, as it led to questions outlined
above. However, it is still important to provide information about the presence or absence of
vapor bubbles in this population and if so, why the depths are still well-constrained.

**As discussed 2 points above regarding melt inclusions in Fo<84 olivines, we have**
**now given further detail about CO₂ bubbles in these inclusions.**

Line 241: This sentence is vague. Please refer to which reservoirs are being replenished,
etc. That is, write this sentence to be more specific to the system at Kīlauea as outlined in
the text – like the SC or the HMM reservoirs that are labeled in figure 4.

We have clarified that this refers to the SC reservoir (Line 384):

“The composition of parental melts supplying the deeper, SC reservoir..”

Line 253: “Just prior to eruption” – what is the evidence that this occurs just prior to eruption?
What is the timescale implied by “just prior?”

As we have not constrained timescales of the final period of transport in this study, we have amended this to read (line 397):

“Prior to eruption, carrier melts scavenge...”

Line 499: Write out dates for the “Ulu period” eruptions. It is difficult for someone unfamiliar with the eruptions in HI to recall this information from the main manuscript.

Amended

Line 508: Write out the number of glass, melt inclusion, and olivine analyses performed. In particular, it is important to note the number of melt inclusions analysed from each of the 4 different samples.

We now add this into the methods section (Lines 703-705) as well as providing this information in table 1.

“We analyse 37, 27, 42, and 20 inclusions respectively from the 1969, 1973, 1974 and 1971 eruptions (see table 1), and ~10 matrix glass chips from each eruption. “

Figure 1 &2: Label/Title 4 dated eruptions from which new data was collected for this study as “Ulu period” or “this study” to better clarify the source of those data.

We now indicate the eruptions measured in this study using bold fonts on Fig. 1 (the map; described in figure captions). We have indicated in the legend for Fig. 2 and 3 which eruptions are measured in this study.

REVIEWERS' COMMENTS:

Reviewer #2 (Remarks to the Author):

The authors have comprehensively addressed the various comments/criticisms of the reviewers. These revisions have been useful in emphasising the important novel aspects of the work and will no doubt aid with the overall impact of the study. I would recommend this version is accepted for publication.

Reviewer #3 (Remarks to the Author):

Overall, I am pleased with the manner in which the authors have thoroughly responded to all reviews. In particular, I feel that the significant changes to the introduction, clarification of the eruption history and data sources, and more detailed explanation of geochemical variations (ME, TE, and CO₂; with useful supplementary figures) have significantly improved the manuscript. In addition, the authors have done a much better job expressing the novelty of their results, while toning down some overstated comments. Based on these changes, I would recommend that the editor accepts the manuscript for publication.